# Prolonged cross-bridge binding triggers muscle dysfunction in a *Drosophila* model of myosin-based hypertrophic cardiomyopathy

William A Kronert[1], Kaylyn M Bell[2], Meera C Viswanathan[3], Girish C Melkani[1], Adriana S Trujillo[1], Alice Huang[2], Anju Melkani[1], Anthony Cammarato[3], Douglas M Swank[2,4]*, Sanford I Bernstein[1]*

[1]Department of Biology, Molecular Biology Institute and Heart Institute, San Diego State University, San Diego, United States; [2]Department of Biology and Center for Biotechnology and Interdisciplinary Studies, Rensselaer Polytechnic Institute, New York, United States; [3]Department of Medicine, Division of Cardiology, Johns Hopkins University, Baltimore, United States; [4]Department of Biomedical Engineering, Rensselaer Polytechnic Institute, New York, United States

**Abstract** K146N is a dominant mutation in human β-cardiac myosin heavy chain, which causes hypertrophic cardiomyopathy. We examined how *Drosophila* muscle responds to this mutation and integratively analyzed the biochemical, physiological and mechanical foundations of the disease. ATPase assays, actin motility, and indirect flight muscle mechanics suggest at least two rate constants of the cross-bridge cycle are altered by the mutation: increased myosin attachment to actin and decreased detachment, yielding prolonged binding. This increases isometric force generation, but also resistive force and work absorption during cyclical contractions, resulting in decreased work, power output, flight ability and degeneration of flight muscle sarcomere morphology. Consistent with prolonged cross-bridge binding serving as the mechanistic basis of the disease and with human phenotypes, *146N/+* hearts are hypercontractile with increased tension generation periods, decreased diastolic/systolic diameters and myofibrillar disarray. This suggests that screening mutated *Drosophila* hearts could rapidly identify hypertrophic cardiomyopathy alleles and treatments.

DOI: https://doi.org/10.7554/eLife.38064.001

*For correspondence:
swankd@rpi.edu (DMS);
sbernstein@sdsu.edu (SIB)

Competing interests: The authors declare that no competing interests exist.

## Introduction

Heritable hypertrophic cardiomyopathy (HCM) is a leading cause of death among young adults, particularly competitive athletes. This heterogeneous and complex disease typically involves asymmetric growth of the heart, interventricular septum thickening, disorganized cellular architecture, diastolic dysfunction, arrhythmias, and an increased risk of sudden cardiac death (*Davis et al., 2016*; *Maron, 2002*; *Maron and Maron, 2013*; *Masarone et al., 2018*). Diastolic dysfunction is characterized by impaired left ventricular relaxation, chamber stiffening, elevated filling pressures, and, consequently, reduced stroke volumes and cardiac output. Enhanced cardiomyocyte contractile activity is a leading hypothesis for the underlying cause of HCM. Increased $Ca^{2+}$-sensitivity of the contractile apparatus as well as heightened function of the myosin molecular motor are potential molecular mechanisms. The over-active contractile apparatus can prolong mechanical tension, which apparently initiates hypertrophy via activation of specific signaling cascades that regulate heart growth (*Davis et al., 2016*). Although therapies exist to treat problematic HCM abnormalities, there is no

**eLife digest** Myosin is a motor protein that drives the contraction of muscles. Filaments made from myosin molecules slide between filaments of another protein called actin, tugging the edges of the muscle cell inwards. To achieve this, part of each motor protein – called the 'head' – grabs hold of actin and uses energy to pull on the filaments.

Small genetic mutations in the gene for myosin can change the shape of the protein. This can change the way that it interacts with actin, altering the molecular machinery that makes muscles contract. In some cases, gene errors can cause the heart muscle wall to thicken, a condition called hypertrophic cardiomyopathy. Mapping the locations of known mutations revealed 'hot spots' on the myosin protein where these errors are likely to cause disease. These include the part of the molecule that swings the myosin heads, and the heads themselves.

It only takes a change to a single letter in the DNA code to thicken the heart wall, but the impact of each possible change is not yet known. Kronert et al. have now genetically modified fruit flies to give them one of the mutations that causes thickening of the heart wall in humans. The mutation, known as K146N, does not appear in one of the well-known 'hot spots'. The experiments revealed that the mutation causes myosin to remain attached to actin for longer than normal. This increased the amount of force the myosin generated, but slowed down actin movement, causing muscle stiffness. This resulted in less power for every cycle of muscle movement, and caused the muscles to degenerate over time. As a result, the flies were less able to use their wings, and their hearts pumped less well.

Hypertrophic cardiomyopathy can cause death in young adults, particularly competitive athletes. Yet studying the disease in humans is challenging. Recreating myosin mutations in fruit flies provides a way to study hypertrophic cardiomyopathy in the laboratory. In the future, extensions to this technique could allow researchers to examine the impact of other mutations. Models like this one could also allow early testing of new drugs or genetic treatments to repair faulty myosin molecules.
DOI: https://doi.org/10.7554/eLife.38064.002

cure. Thus, it is critical to better understand both the disease-causing mutations and the mechanism by which they produce HCM.

Over 1400 point mutations in sarcomeric proteins cause HCM (*Maron and Maron, 2013*), with the largest number (>300) found in the β-cardiac myosin heavy chain. This myosin II molecule forms sarcomere thick filaments and serves as a molecular motor to drive ATP-dependent movement along actin-containing thin filaments. Mapping the locations of the mutations onto a three-dimensional structure of myosin reveals several hot spots. These include the converter domain (*Colegrave and Peckham, 2014*) and the recently described 'myosin mesa' (*Homburger et al., 2016*). The converter guides the myosin lever arm in its rotation during the power stroke that generates force and motion (*Mesentean et al., 2007*). The myosin mesa has been implicated in enabling formation of an interacting head motif (*Trivedi et al., 2018*), wherein the heads of the dimeric molecule fold together to dampen motor function (*Woodhead et al., 2005*). Several molecular and physiological studies suggest that enhanced ATPase, in vitro actin sliding speed and muscle contractile force and/or velocity correlate with many, but not all, of the HCM-causing myosin mutations (*Adhikari et al., 2016*; *Moore et al., 2012*; *Sommese et al., 2013*; *Spudich, 2014*). For the domains cited above, this could occur via activation of the lever arm through enhanced converter function or stimulation of myosin cross-bridge initiation through inhibiting the formation of the interacting head motif. The mechanism by which increased contractility results in development of HCM phenotypes remains an active area of investigation (*Davis et al., 2016*; *Garfinkel et al., 2018*).

In this communication, we model a human myosin HCM mutation in *Drosophila melanogaster*, which provides a highly integrative approach that yields unique insights into the disease. The *Drosophila* heart, although a relatively simple tube-shaped structure, shares numerous gene expression patterns, as well as developmental and functional properties, with the human heart (*Bier and Bodmer, 2004*). Further, the fly system allows evaluation of the effects of a mutation upon isolated myosin (*Swank et al., 2000*), as well as on the myofibrillar structure and physiological function of both skeletal (*Maughan and Vigoreaux, 1999*) and cardiac muscles (*Ocorr et al., 2014*). A major

advantage of *Drosophila* is its simplified genetics, in that one *Mhc* gene gives rise to all myosin iso-forms through alternative RNA splicing (*George et al., 1989*). Hence compensation by myosin multi-gene family members that can mask mutant protein effects does not occur. Further, myosin expression levels and tissue-specificity can be controlled by combining *Mhc* transgenes with lines containing *Mhc* alleles that specifically knock out myosin heavy chain expression in the indirect flight muscles (*Collier et al., 1990*) or in all muscle types (*O'Donnell and Bernstein, 1988*). Previously, using this approach, we showed that a mutant myosin with enhanced ATPase activity yielded dis-rupted skeletal muscle structure and function and restricted the *Drosophila* cardiac tube to reduce its output (*Cammarato et al., 2008*). This indicates that *Drosophila* models could yield insights into the basis of HCM through assessing human mutations that cause overactive myosin.

For the current study, we produced transgenic flies expressing the *Drosophila* equivalent of the K146N human HCM myosin mutation. This mutation was identified in the family of an adult patient with increased left ventricular wall thickness in the absence of hypertension (*Ingles et al., 2005*) and as a spontaneous mutation in a child who displayed left ventricular hypertrophy (*Morita et al., 2008*). Residue 146 is particularly interesting in that it does not map to a well-studied HCM hot spot. Instead, it maps near the N-terminus of the protein, which has few HCM-causing mutations (*Colegrave and Peckham, 2014*). However, mutational analysis in *Dictyostelium* (*Ruppel et al., 1994*) and a domain swap study in *Drosophila* (*Swank et al., 2003*) suggest that effects on the mechanochemical cycle of myosin might be expected from alterations to the N-terminal region of myosin II molecules. Our molecular modeling predicts that mutation of residue 146 of muscle myosin reduces its interaction with the myosin lever arm during the pre-power stroke state. The myosin S1 head domain is in a cocked position during this state, so that it is prepared to drive the power stroke and muscle contraction upon actin binding during the mechanochemical cycle. We found that the primary result of the 146N mutation is that myosin spends increased time in strongly bound states of the cross-bridge cycle. This prolonged binding of myosin to actin increases force generation, but slows actin motility, decreases cyclical muscle power generation and reduces flight ability. Higher force production and stiffness also cause progressive skeletal and cardiac muscle structural degener-ation, and a restricted cardiac phenotype with diastolic dysfunction. Some of our observations, such as the prolonged periods of systolic tension as well as increased ATPase rates, indicate this HCM mutation causes hyperdynamic contractile properties. However, our use of the integrative approach in the *Drosophila* system shows that the mutant phenotype is complex, with reduced functions for a number of other parameters. This emphasizes the need for a comprehensive analysis, from the molecular to organismal levels, which has allowed us to provide key insights into defining the molec-ular mechanism behind myosin-based HCM.

## Results

### Molecular modeling of myosin residue 146 and predicted effects of the HCM mutation

We determined the location and interactions for the amino acid residue corresponding to human K146, which in its mutant form (K146N) causes HCM. The *Drosophila* indirect flight muscle (IFM) myosin heavy chain sequence was modeled onto the scallop muscle myosin II crystal structure during the pre-power stroke state (PDB 1QVI) and the actin-detached post-power stroke state (PDB 1KK8) (*Figure 1A and B*). Both scallop and *Drosophila* substitute an identically charged arginine residue for lysine at residue 146, as do some human muscle myosins (*Rossi et al., 2010*) (for clarity of com-parison, the human β-cardiac myosin numbering system is used throughout). The location of the modeled *Drosophila* residue, which is at the surface of the motor domain of the molecule, is shown in magenta. In the pre-power stroke state, this residue is in close proximity to the lever arm (*Figure 1A*), a domain that rotates to generate movement and force during muscle contraction. As a result of configuration changes during the mechanochemical cycle, the N-terminus and R146 are dis-tant from the lever arm in the post-power stroke state (*Figure 1B*). The *Drosophila*, scallop and human β-cardiac myosins show strong sequence similarity in the R146 region and also in the lever arm interacting region (*Figure 1B* inset).

Analysis of the pre-power stroke model reveals specific charge interactions with residue 146 (*Figure 1C*). There is a 3.2 Å contact distance between positively charged R146 and negatively

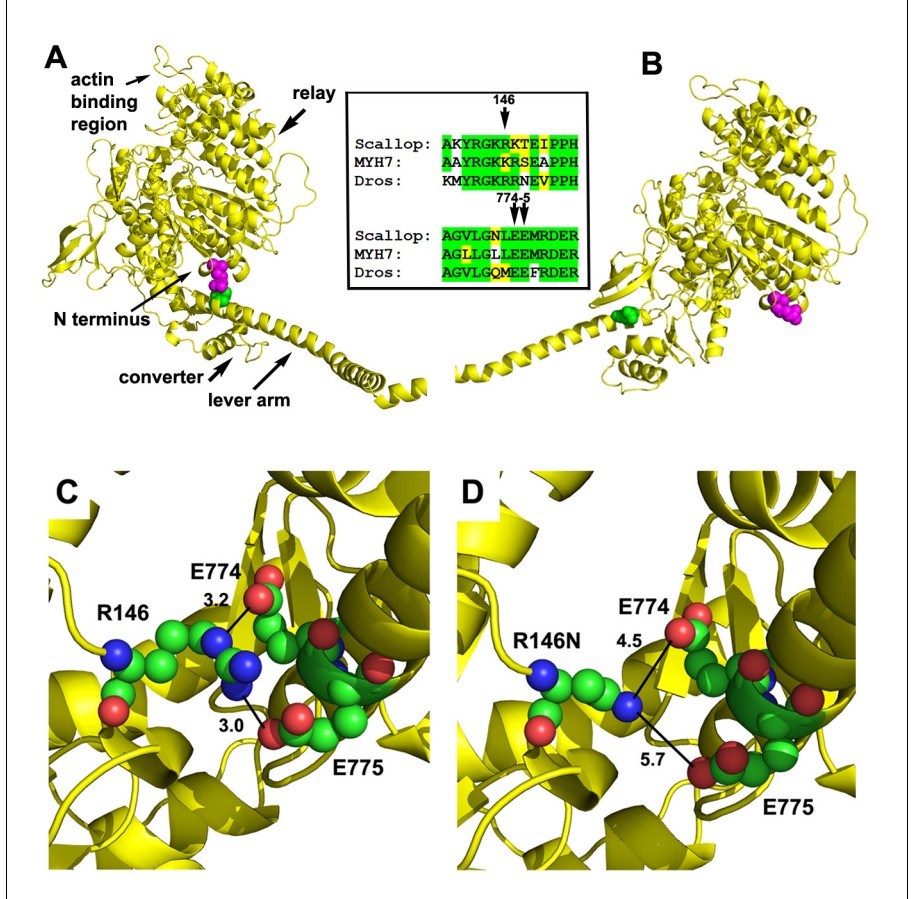

**Figure 1.** Location and interactions of R146 within the myosin molecule during pre- and post-power stroke states. (A) Scallop myosin S1 in the pre-power stroke state (1QVI) was used as a template to model *Drosophila* IFM S1. R146, the positively charged residue involved in HCM is depicted with magenta spheres. In this state, the lever arm is in close proximity to R146, with the site of interaction colored green. The inset shows the amino acid context for R146 (top) and its interaction sites on the lever arm (bottom) in scallop, *Drosophila* and human β-cardiac myosin. Identical (green) and conserved (yellow) residues demonstrate a high degree of sequence similarity. (B) Scallop myosin S1 in the post-power stroke state (1KK8) was used as a template to model *Drosophila* IFM S1. In this state, R146 is distant from the lever arm interaction site (green). (C) Close-up view of potential interactions of R146 in modeled *Drosophila* IFM S1 in the pre-power stroke state. R146 displays charge interaction with E774 with a contact distance of 3.2 Å. A second charge interaction occurs between R146 and E775 with a contact distance of 3.0 Å. The close contact distances permit the formation of salt bridges between the positively charged R146 and the negatively charged E774 and E775 residues. (D) The R146N mutation eliminates the close contacts with E774 and E775 during the pre-power stroke state. R146N to E774 distance is 4.5 Å and R146N to E775 distance is 5.7 Å.

DOI: https://doi.org/10.7554/eLife.38064.003

The following figure supplement is available for figure 1:

**Figure supplement 1.** Analysis of the K146 residue in mammalian cardiac myosin in the pre-power stroke state.

DOI: https://doi.org/10.7554/eLife.38064.004

charged E774, as well as a 3.0 Å contact distance between R146 and E775. These close contact distances permit the formation of salt bridges with E774 and E775 residues in the lever arm of the molecule. In a model with HCM mutant residue R146N, the close contact and salt bridges with E774 and E775 are eliminated, with distances increased to 4.5 and 5.7 Å, respectively (*Figure 1D*). This could decrease the stability of the pre-power stroke state and alter rates of conformational changes required for the actin-myosin mechanochemical cycle to proceed normally. A similar disruption may occur for the human β-cardiac mutant myosin as well (*Figure 1—figure supplement 1*).

## Production and verification of R146N transgenic lines

To study the effects of the R146N mutation within myosin and *Drosophila* muscle, we constructed a mutant myosin transgene and obtained 29 transgenic lines by *P* element-mediated transformation. Three independent transgenic inserts that mapped to the third chromosome (*PwMhcR146N-11*, *PwMhcR146N-15*, *PwMhcR146N-28*; abbreviated hereafter with *PwMhc* deleted) were crossed into the $Mhc^{10}$ background, which is null for myosin heavy chain in IFM (*Collier et al., 1990*). RT-PCR of RNA isolated from dissected IFM verified that each transgenic line expresses only the mutant myosin in the IFM and that the normal alternative exon splicing pattern is not disrupted (see Materials and methods). SDS-PAGE analysis confirmed that upper thoraces of mutant 2-day-old adult female flies from lines 11, 15 and 28 express normalized ratios of myosin to actin ($1.06 \pm 0.06$, $0.99 \pm 0.02$, $0.98 \pm 0.01$, respectively) that are essentially identical to the wild-type control transgenic line *PwMhc2* ($1.00 \pm 0.04$).

## R146N doubles basal myosin ATPase activities, but slows in vitro motility

To explore the impact of the R146N mutation on the mechanochemical cycle of myosin, we assessed the steady-state ATPase parameters of myosin isolated from the IFM of *R146N* homozygotes (*Figure 2A–E*). Both Ca-ATPase ($16.64 \pm 1.87$ vs. $9.51 \pm 1.08$ sec$^{-1}$) and basal Mg-ATPase ($0.62 \pm 0.14$ vs. $0.25 \pm 0.04$ sec$^{-1}$) rates of R146N myosin increased significantly, about two-fold compared to wild-type myosin (*Figure 2A,B*). While $Ca^{2+}$ typically yields higher in vitro ATPase activities, $Mg^{2+}$ is the physiologically relevant cation that is employed for actin stimulation. We found that actin-stimulated Mg-ATPase activity ($V_{max}$) (*Figure 2C*) was not significantly different from the control ($1.57 \pm 0.27$ vs. $1.62 \pm 0.14$ sec$^{-1}$). The $K_m$ of actin concentration relative to ATPase activity was significantly increased, indicating that a 35% higher actin concentration is required to reach 50% $V_{max}$ ($0.42 \pm 0.06$ vs. $0.31 \pm 0.04$ µM) (*Figure 2D*). Further, the mutant's catalytic efficiency (defined as $V_{max}/K_m$) was significantly lower than that of the control ($3.80 \pm 0.60$ vs. $5.34 \pm 1.07$ sec$^{-1}$/µM) (*Figure 2E*). Analysis of in vitro actin-sliding velocity data showed that the mutant myosin possesses an ~two fold reduction in its ability to drive actin filament sliding compared to myosin isolated from control IFM ($3.32 \pm 0.21$ vs. $6.71 \pm 0.17$ µm/s) (*Figure 2F*). Thus the major effects of the R146N mutation on ATPase and in vitro motility are an enhancement of basal Mg- and Ca-ATPase rates and a decrease in actin filament sliding velocity.

## R146N myosin disrupts IFM myofibril stability

To determine if the R146N mutation affects muscle structure and stability, we employed transmission electron microscopy to examine IFMs of homozygous female flies at late-pupal stage as well as fibers from 2-hr-old, 2-day-old and 1-week-old adults. Transverse and longitudinal sections of *R146N* late-stage pupae and 2-hr-old adults (*Figure 3E,F*) showed normal sarcomeres containing double hexagonal arrays of thick and thin filaments, which are indistinguishable from those of the *PwMhc2* homozygous control line at the same developmental stages (*Figure 3A,B*). Transverse and longitudinal sections of *R146N* 2-day-old adults showed minor disruptions of thick and thin filament packing (*Figure 3G*) compared to the control (*Figure 3C*), with even more modest disruption seen in *R146N/+* heterozygotes (*Figure 3—figure supplement 1*). Transverse and longitudinal sections of *R146N* 7-day-old adults showed more disorder in myofibril morphology, with gaps in the hexagonal packing of thick and thin filaments and disruption in the Z- and M-lines (*Figure 3H*) compared to the control (*Figure 3D*). Hence R146N mutant myosin is capable of contributing to the formation of normal myofibrillar structures that begin to deteriorate at 2 days into adulthood, presumably as a result of mechanical stress during use. We minimized any influence of structural changes in *R146N* adult IFMs on our muscle mechanical assays by employing fibers from 2-hr-old flies for these assays. We also assessed heterozygous organisms (*R146N/+*) to determine whether the dominant nature of the mutation in humans is mirrored in *Drosophila*.

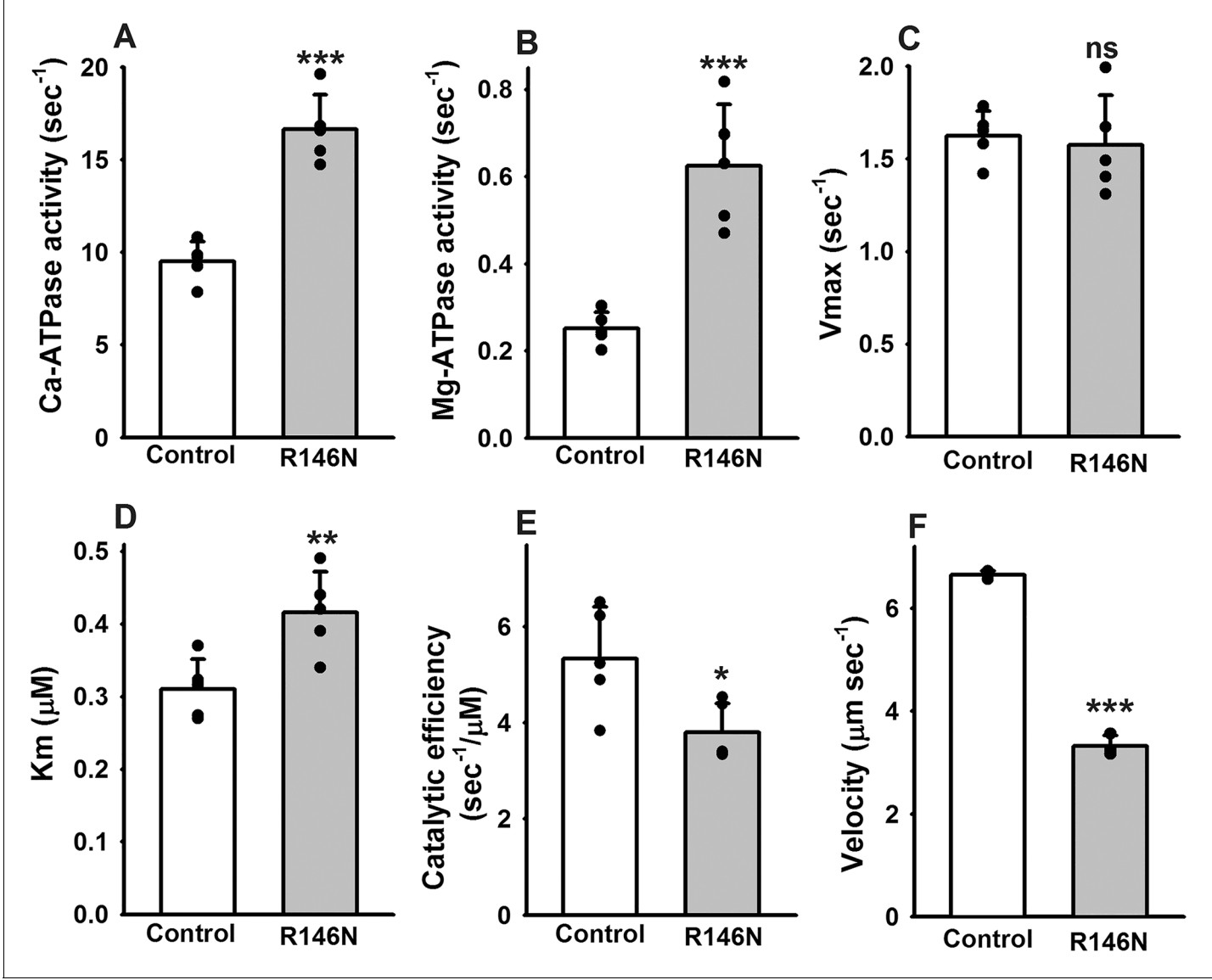

**Figure 2.** Effects of R146N on myosin enzymatic activity and in vitro actin sliding velocity. Myosin isolated from IFMs of wild-type transgenic control (*PwMhc2*) and *R146N* lines was assessed for ATPase parameters (N = 5): (A) Ca-ATPase activity, (B) basal Mg-ATPase activity (C) actin-activated Mg-ATPase activity ($V_{max}$), (D) actin affinity relative to ATPase ($K_m$) and (E) catalytic efficiency ($V_{max}/K_m$). Actin sliding velocity (F) was determined using the in vitro motility assay, with at least 30 filaments tracked per myosin preparation (N = 3). For all parameters, mean ± S.D. values are given. These were assessed for statistically significant differences by Student's t-tests. *p<0.05, **p<0.01, ***p<0.001, ns, not significant.
DOI: https://doi.org/10.7554/eLife.38064.005

## Muscle mechanical assays show that R146N myosin influences work, power and cross-bridge cycle rate constants

### R146N reduces work and power output

We employed two muscle assays to determine the influence of the R146N mutation on IFM fiber mechanical properties (*Figures 4* and *5*). The first, small amplitude sinusoidal analysis, involves imposing a 0.125% sinusoidal length change on an active muscle fiber and recording the resultant fiber force output. By measuring the amplitude and phase shift of the force trace relative to the length trace, muscle properties such as muscle stiffness, power, work, and cross-bridge kinetics can be elucidated (*Figure 5—figure supplement 1*) (*Kawai and Brandt, 1980*; *Swank, 2012*). The second assay, the work loop technique, is a more physiological measure of work and power because

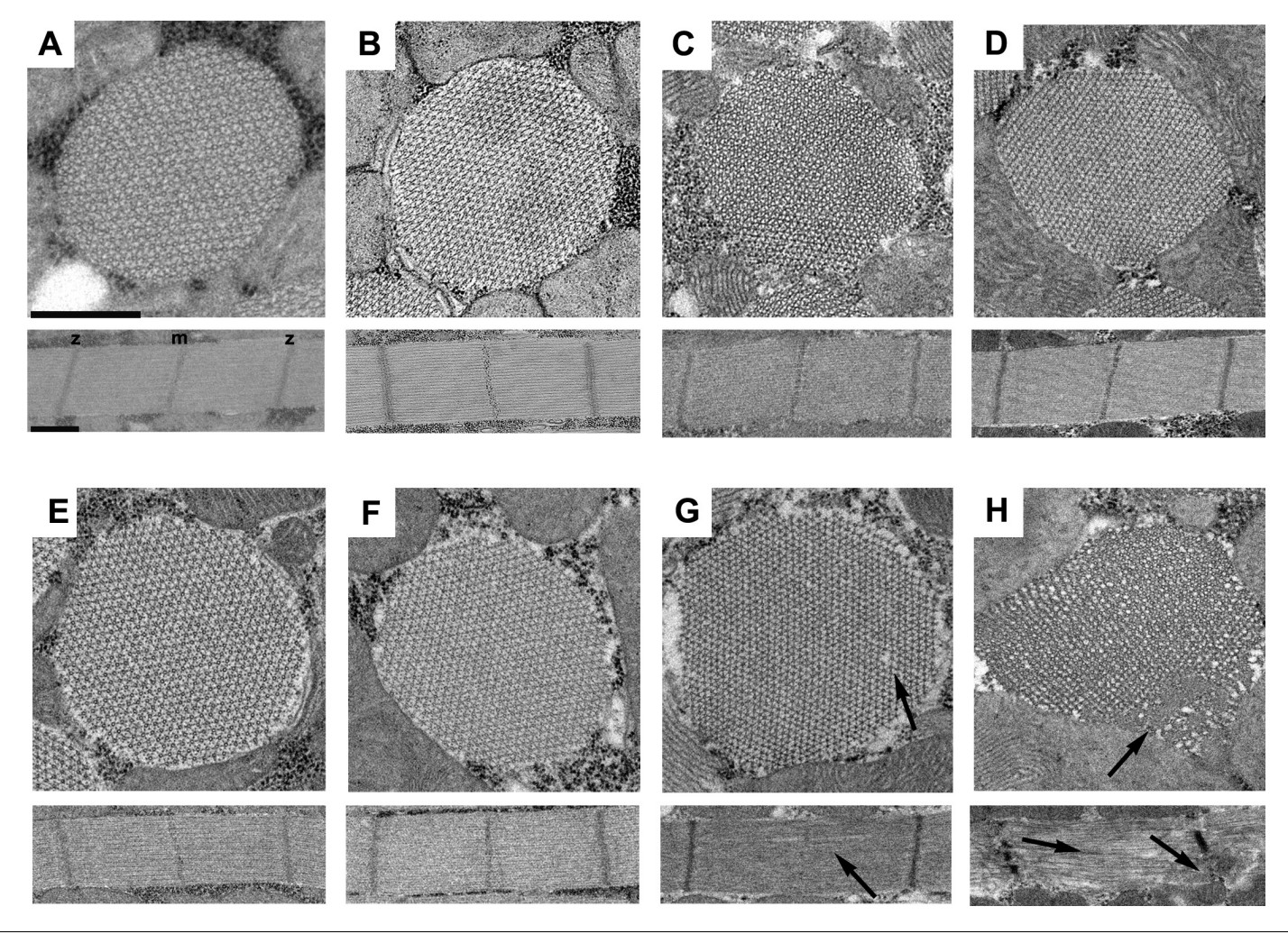

**Figure 3.** The R146N myosin mutation disrupts IFM myofibril stability. Transverse and longitudinal sections are shown from the homozygous control line, *PwMhc2* (A–D), and *R146N* homozygotes (E–H), for four different developmental stages: (A, E) late-stage pupae, (B, F) 2-hr-old adults, (C, G) 2-day-old adults, (D, H) 7-day-old adults. Each panel shows one myofibril that is representative of the population for that stage of development. Thick and thin filaments are arrayed in a normal double hexagonal pattern and well formed Z- and M-lines are observed in the sarcomeres at all stages in control organisms and at the late pupal and 2-hr-old adult stages for R146N. Mild disorder in the hexagonal packing of thick and thin filaments and some M-line disruption of sarcomeres (arrows) is observed in 2-day-old mutant adults (G). The structure of the mutant further deteriorates (arrows) in seven-day-old adults (H), with myofilament packing defects and strong disruption of both M- and Z-lines (m, m-line, z, z-line). All three *R146N* lines yielded similar phenotypes. Scale bars, 0.5 μm. Full genotypes are shown in parentheses: *PwMhc2* homozygous control (P{PwMhc2}/P{PwMhc2}; Mhc$^{10}$/Mhc$^{10}$); R146N homozygote (Mhc$^{10}$/Mhc$^{10}$; P{R146N}/P{R146N}).

DOI: https://doi.org/10.7554/eLife.38064.006

The following figure supplement is available for figure 3:

**Figure supplement 1.** The *R146N* myosin mutation acts in a dominant fashion to cause minor disruption in IFM myofibril stability.

DOI: https://doi.org/10.7554/eLife.38064.007

longer muscle length changes are used. Work and power are calculated by plotting the imposed length change versus force and integrating the area inside the resulting loop (*Josephson, 1985*). Sinusoidal analysis revealed that fibers from homozygotes (*R146N/R146N*) and heterozygotes (*R146N/+*) showed decreased IFM fiber power generation over the frequency ranges 40–210 Hz and 40–100 Hz, respectively (*Figure 4A*). The average optimal power generation by homozygous fibers was 43% lower than for the *PwMhc2* control line, but there was not a significant decrease in maximum power generation by the heterozygous fibers (*Table 1*). The decrease in power from homozygous fibers appears to be primarily caused by a decrease in net work generation, rather than an

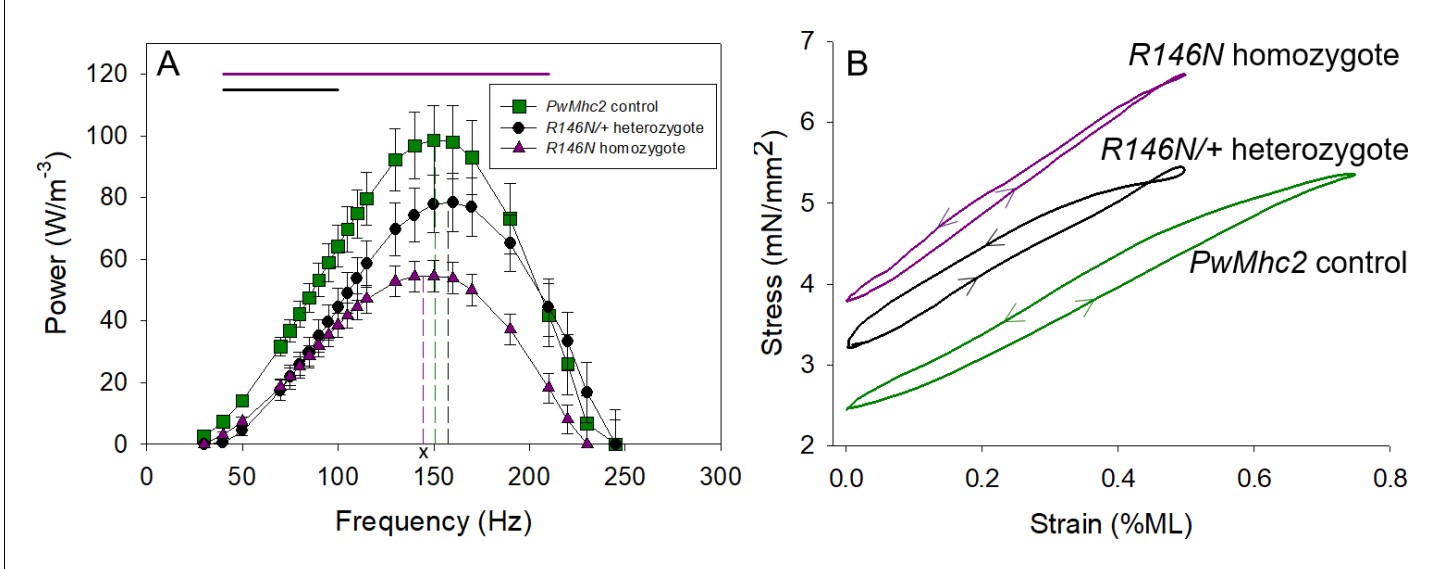

**Figure 4.** Power and work are decreased by the R146N mutation. (**A**) Muscle power was measured by small amplitude, 0.125% muscle length (ML), sinusoidal analysis over a range of oscillation frequencies. Control fibers (*PwMhc2* homozygotes) are represented by green squares, heterozygous fibers (*R146N-15/+*) by black circles and homozygous fibers (*R146N-15/R146N-15*) by purple triangles. Dashed lines in green, black and purple represent the respective fibers' $f_{max}$. ˣ Significant difference from *R146N-15/+* heterozygotes at $p<0.05$. Bars indicate statistical significance from control, $p<0.05$, Student's t-test. N = 12 for control and heterozygous fibers, 16 for homozygous fibers. (**B**) Work and power were measured using the work loop technique, which allows for variation in ML oscillation. In this representative example, the fibers were subjected to different ML amplitudes and oscillation frequencies until conditions were found that generated maximum power. The area within the loop is the amount of net work generated per cycle, and power is calculated by multiplying net-work times oscillation frequency. The control fiber (green) has a larger loop area than those of the heterozygous (black) or homozygous (purple) mutant fibers. Arrows show the counter-clockwise direction of the work loops indicating net work production rather than net work absorption. The mutant fibers perform best at 0.5% ML, 125 Hz and 150 Hz, for the *R146N-15* homozygous and *R146N-15/+* heterozygous fibers, respectively, while the control fiber produced maximum power at 0.75% ML, 125 Hz. Full genotypes are shown in parentheses: *PwMhc2* homozygote control ($P\{PwMhc2\}/P\{PwMhc2\}$; $Mhc^{10}/Mhc^{10}$); *R146N-15/+* heterozygote ($Mhc^{10}/+$; $P\{R146N-15\}$); *R146N-15* homozygote ($Mhc^{10}/Mhc^{10}$; $P\{R146N-15\}/P\{R146N-15\}$).

DOI: https://doi.org/10.7554/eLife.38064.008

overall slowing of average muscle kinetics (power = work*frequency). This is suggested by homozygous mutant fibers showing the same percent drop in maximum work production as maximum power production (both 40%) and because there were not significant differences in the frequencies at which maximum power ($f_{max}$) and the frequency at which maximum work ($f_{Wmax}$) were generated compared to control values (**Table 1**). However, the frequency of maximal power ($f_{max}$) and the frequency of maximal work ($f_{Wmax}$) values for *R146N/+* heterozygous fibers were significantly greater than those from *R146N* homozygous fibers, suggesting some influence of the mutation on the underlying myosin kinetics.

The work loop technique takes advantage of the fact that muscle shows a unique response compared to other materials when measuring its stress during lengthening and shortening contraction cycles (**Josephson, 1985**). While most materials produce a clockwise work loop indicating that work is absorbed by the material, an active muscle is able to generate work and power, as indicated by a counter-clockwise work loop (arrows in **Figure 4B**). We first optimized muscle length change amplitude and frequency of oscillation until parameters were found that produced maximum power for each fiber type. Even when optimized, fibers from both the *R146N/+* heterozygotes and the *R146N* homozygotes produced about 60% less work and 43% and 65% less power, respectively, than control fibers (**Figure 4B**, **Table 2**). We next measured the mutant fibers' ability to generate power when tested under the control fibers' typical optimal power producing parameters (**Supplementary file 1**). Under these parameters, power and work generation decreased further: 60% for heterozygotes and 90% for homozygotes compared to that generated under their optimized parameters (compare **Table 2** to **Supplementary file 1**). This is likely due to the increase in percent muscle length change from 0.5% to 0.75%. In both cases, the net work and power output

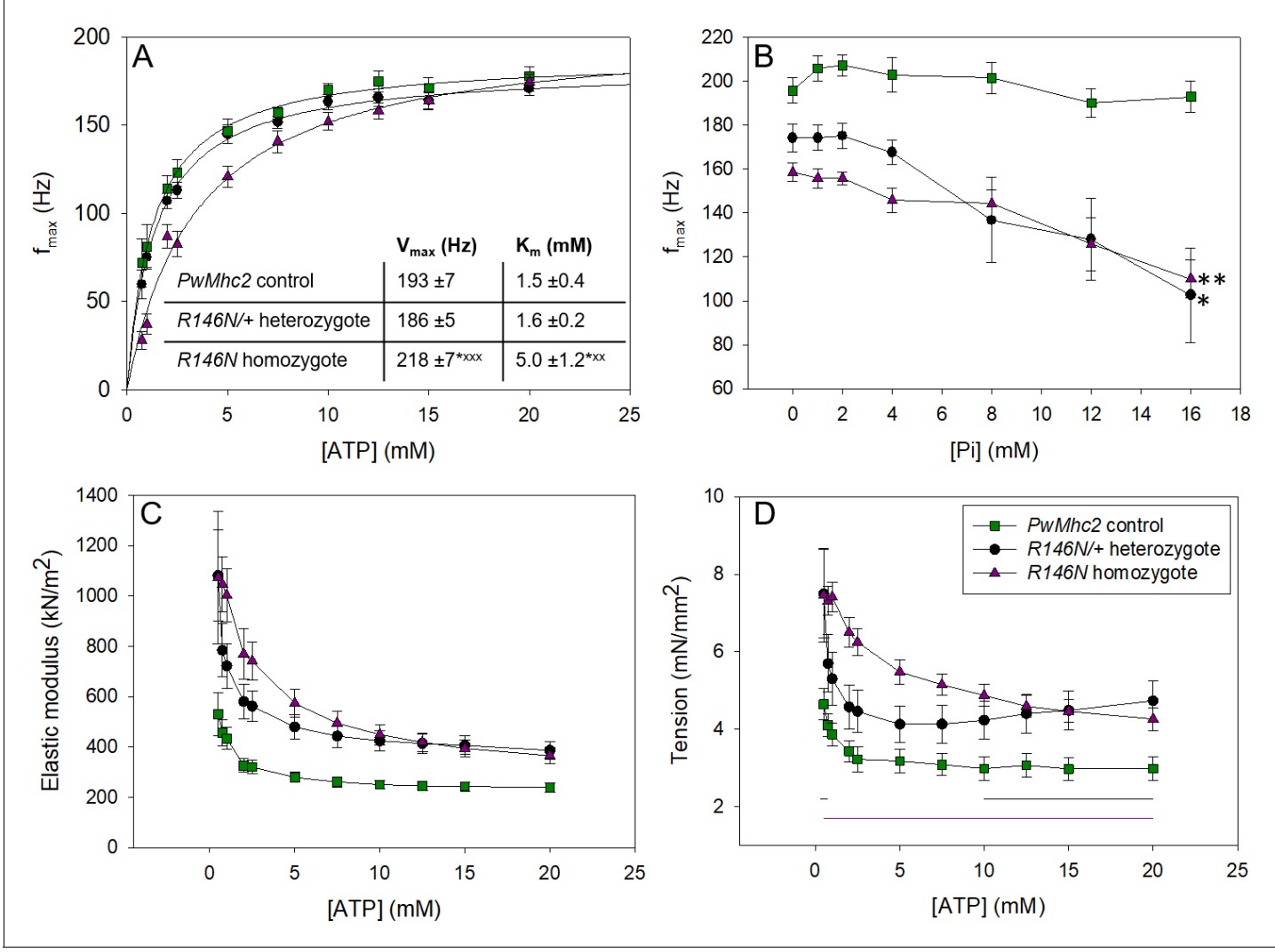

**Figure 5.** Varying [ATP] and [Pi] reveals information about cross-bridge kinetics. (**A**) The frequency at which small amplitude sinusoidal power was maximal, $f_{max}$, was measured for a range of ATP concentrations for control (*PwMhc2* homozygotes), heterozygous (*R146N-15/+*) and homozygous (*R146N-15/R146N-15*) fibers. The curves were fit with the Michaelis-Menten equation and $V_{max}$ and $K_m$ values determined (see inset). Student's t-test with $p < 0.05$ significantly different from control (*$p < 0.05$, **$p < 0.01$, ***$p < 0.001$) or heterozygote (x$p < 0.05$, xx$p < 0.01$, xxx$p < 0.001$). (**B**) $f_{max}$ relative to inorganic phosphate concentration. $f_{max}$ decreased with increasing [Pi], suggesting phosphate is competing with ATP for the rigor state. * indicates statistically significant difference from 0 mM Pi (Student's t-test with $p < 0.05$ significantly different (*$p < 0.05$, **$p < 0.01$). Linear regression analysis showed mutants had a steeper decline of $f_{max}$ and larger negative slopes than the control. Seven control, 12 heterozygous, and seven homozygous fibers were tested. (**C**) Change in active stiffness with increasing ATP concentration. In-phase stiffness (elastic modulus) was measured at pCa 5.0 at 500 Hz ML oscillation using small amplitude sinusoidal analysis. All *R146N* stiffness values are statistically greater than the control's by a Student's t-test at $p < 0.05$. (**D**) Active isometric tension generation as a function of ATP concentration at pCa 5.0. For the ATP study, N = 12 control, 12 heterozygous and 14 homozygous fibers. Black and purple horizontal lines indicate [ATP] where heterozygous and homozygous tension values are statistically greater than control's (t-test, $p < 0.05$). Full genotypes are shown in parentheses: *PwMhc2* homozygote control (*P{PwMhc2}/P{PwMhc2}; Mhc[10]/Mhc[10]*); *R146N-15/+* heterozygote (*Mhc[10]/+; P{R146N-15}*); *R146N-15* homozygote (*Mhc[10]/Mhc[10]; P{R146N-15}/P{R146N-15}*).

DOI: https://doi.org/10.7554/eLife.38064.009

The following figure supplement is available for figure 5:

**Figure supplement 1.** Acto-myosin cross-bridge cycle scheme.

DOI: https://doi.org/10.7554/eLife.38064.010

**Table 1.** Small amplitude power, work, and isometric tension.

Means ± S.E.M are reported. Student's t-test with p<0.05 significantly different from control (*p<0.05, **p<0.01, ***p<0.001) or heterozygote ($^x$p<0.05, $^{xx}$p<0.01, $^{xxx}$p<0.001). $f_{max}$: the frequency at which maximum power was generated, $f_{Wmax}$: frequency at which maximum work was generated as determined using small amplitude sinusoidal analysis. Passive tension was measured at pCa 8.0 and active at pCa 5.0. N = 12 fibers for control and mutant heterozygotes; N = 16 for mutant homozygotes. Full genotypes are shown in parentheses: *PwMhc2* control (P{PwMhc2}/P{PwMhc2}; *Mhc$^{10}$/Mhc$^{10}$*); *R146N-15/+* heterozygote (*Mhc$^{10}$/+; P{R146N-15}*); *R146N-15* homozygote (*Mhc$^{10}$/Mhc$^{10}$; P{R146N-15}/P{R146N-15}*).

| Line | Power (W/m$^3$) | $f_{max}$ (Hz) | Work (J/m$^3$) | $f_{Wmax}$ (Hz) | Passive Tension | Active Tension |
|---|---|---|---|---|---|---|
| *PwMhc2* control | 100 ± 11 | 151 ± 3 | 0.71 ± 0.08 | 128 ± 2 | 1.94 ± 0.30 | 3.07 ± 0.32 |
| *R146N-15/+* heterozygote | 81 ± 10 | 158 ± 4 | 0.54 ± 0.07 | 136 ± 4 | 3.16 ± 0.40* | 4.40 ± 0.51* |
| *R146N-15* homozygote | 57 ± 5***$^x$ | 145 ± 4$^x$ | 0.41 ± 0.04** | 122 ± 4$^x$ | 2.73 ± 0.32 | 4.59 ± 0.29** |

DOI: https://doi.org/10.7554/eLife.38064.011

decrease resulted from the ratio of work generated to work absorbed being lowered by the mutation, resulting in less net (useful) work and power being produced during the IFM's cyclical contractions. This was the case even when mutant work generated was higher than the control work generated (homozygotes in *Supplementary file 1*).

## R146N alters cross-bridge cycle rate constants

To gain additional insight into possible changes in cross-bridge kinetics resulting from the R146N mutation, we derived apparent muscle rate constants by fitting Nyquist curves generated by sinusoidal analysis using a complex modulus equation (*Figure 5—figure supplement 1*) (*Kawai and Brandt, 1980*; *Swank, 2012*). While amplitudes A, B and C from the mutant showed no differences compared to control values, there were significant changes in rate constants 2πb and 2πc (*Table 3*). 2πb values are influenced largely by steps associated with myosin attachment to actin and the power stroke, while 2πc values are influenced principally by steps associated with detachment of myosin from actin. Under maximum power generating conditions, 2πb was 30% greater for *R146N/+* heterozygote fibers compared to control fibers, while 2πc was 13% lower for *R146N* homozygote fibers than control fibers (*Table 3*). These data suggest that the presence of the mutation increases the speed of at least one step associated with attachment rate and the power stroke, but decreases the speed of at least one step associated with myosin detachment rate. According to the interpretation of *Palmer et al. (2007)*, this also suggests that the myosin spends a longer time in strongly bound cross-bridge states.

To determine if the slower detachment rate could be caused by decreased ATP-induced detachment of myosin from actin, we varied ATP concentration while performing sinusoidal analysis. The most informative result was observed when plotting [ATP] vs. $f_{max}$ (*Figure 5A*). By fitting these data with a Michaelis-Menten curve, we found that the fibers from *R146N* homozygotes had a threefold

**Table 2.** Optimized work loops.

Muscle length (ML) change amplitude and frequency of large amplitude sinudoidal oscillations were varied until maximum power generating parameters were obtained. $f_{Pmax}$: the frequency at which maximum power was generated as determined using the work loop technique. Student's t-test with p<0.05 significantly different from control (*p<0.05, **p<0.01, ***p<0.001) or heterozygote ($^x$p<0.05, $^{xx}$p<0.01, $^{xxx}$p<0.001). N = 12 for control and mutant heterozygotes; N = 16 for mutant homozygotes. Full genotypes given in legend to *Table 1*.

| Line | Power (W/m$^3$) | Net work (J/m$^3$) | Work Gen. (J/m$^3$) | Work abs. (J/m$^3$) | % ML Amplitude | $f_{Pmax}$ (Hz) |
|---|---|---|---|---|---|---|
| *PwMhc2* control | 240 ± 34 | 2.18 ± 0.3 | 36.16 ± 4.19 | 33.98 ± 3.95 | 0.79 ± 0.06 | 115 ± 8 |
| *R146N-15/+* heterozygote | 137 ± 20* | 0.93 ± 0.1*** | 21.92 ± 2.70** | 20.99 ± 2.59** | 0.5 ± 0.03*** | 150 ± 5** |
| *R146N-15* homozygote | 85 ± 7***$^x$ | 0.65 ± 0.06*** | 22.50 ± 1.51** | 21.84 ± 1.46** | 0.52 ± 0.03*** | 130 ± 3$^{xx}$ |

DOI: https://doi.org/10.7554/eLife.38064.012

**Table 3.** Muscle apparent rate constants derived from sinusoidal analysis (*Kawai and Brandt, 1980*).
Means ± S.E.M are reported. Student's t-test with p<0.05 significantly different from control (*p<0.05, **p<0.01, ***p<0.001). N = 12 for control and mutant heterozygotes; N = 16 for mutant homozygotes (see *Figure 5—figure supplement 1* for an explanation of how the apparent rate constants relate to the cross-bridge cycle). Full genotypes given in legend to *Table 1*.

| Line | A | B | $2\pi b$ (s$^{-1}$) | C | $2\pi c$ (s$^{-1}$) |
|---|---|---|---|---|---|
| *PwMhc2* control | 339 ± 23 | 741 ± 73 | 1036 ± 35 | 923 ± 76 | 5376 ± 169 |
| *R146N-15/+* heterozygote | 305 ± 33 | 832 ± 75 | 1343 ± 52*** | 1009 ± 85 | 4928 ± 262 |
| *R146N-15* homozygote | 304 ± 33 | 751 ± 87 | 1224 ± 80 | 934 ± 93 | 4671 ± 232* |

DOI: https://doi.org/10.7554/eLife.38064.013

larger $K_m$ value for $f_{max}$ than the control, 5.0 ± 1.2 mM vs. 1.5 ± 0.4 mM, respectively, suggesting that the mutant fibers have decreased affinity for ATP. The $K_m$ value of fibers from *R146N/+* heterozygotes, 1.6 ± 0.2 mM, was not statistically different from the control. Results obtained from varying phosphate concentration also suggested changes to cross-bridge kinetics. The control fibers showed no change in $f_{max}$ (*Figure 5B*) with increasing [Pi], as is typical of IFM fibers (*Swank et al., 2006*). However, fibers from both the homozygous and heterozygous mutants showed that $f_{max}$ decreased as [Pi] increased, with plots showing significantly steeper slopes than the control. This decrease is typically attributed to phosphate competing with ATP for the rigor state (*Pate and Cooke, 1989*), again suggesting lower affinity of the mutant fibers for ATP.

If R146N myosin spends a longer time in strongly bound states than control myosin, one would expect increased stiffness and isometric tension generation in muscle fibers. We indeed found that fiber active stiffness, measured by oscillating fibers at 500 Hz, was greater than control levels at all ATP concentrations tested, for fibers from both *R146N/+* heterozygous and *R146N* homozygous organisms (*Figure 5C*). Likewise, fibers from both heterozygous and homozygous flies showed total calcium-activated isometric tension levels that were at least 1.5-fold greater than the control at all ATP concentrations tested (*Figure 5D*). Interestingly, analysis of passive (relaxed) isometric tension at pCa 8.0 revealed that some of the increase in isometric tension is due to an increase in passive tension, particularly for heterozygous fibers (*Table 1*). This suggests that more cross-bridges were binding to actin at low calcium concentrations in the mutant fibers compared to control. Based on the muscle mechanics results, it appears that the R146N mutation changes at least two rates of the cross-bridge cycle. The mutation increases a cross-bridge rate associated with myosin attachment to actin and/or the power stroke, and decreases a rate associated with ATP-induced myosin detachment from actin.

## R146N myosin decreases flight ability

We assessed the effects of R146N myosin on IFM function by testing flight ability. Line 15 *R146N* homozygotes exhibited ~50% and ~70% decreases in flight index at 15°C (temperature employed for flight muscle mechanics) and 22°C (ambient temperature), respectively, compared to the control line (*Table 4*). The flight index of *R146N/+* heterozygotes was significantly higher than that of

**Table 4.** Flight ability and wing beat frequency (WBF) of *R146N* mutants.
Adult female flies were aged for 2 days prior to flight testing. Transgenic flies were then assayed for the ability to fly up (U), horizontal (H), down (D) or not at all (N). Flight index = 6 U/T+4 H/T+2 D/T+0 N/T; T is the total number of flies tested (given in parentheses). Flight index and WBF values are mean ±S.E.M. Student's t-test with p<0.05 significantly different from 2-day-old *PwMhc2* (*p<0.05, **p<0.01, ***p<0.001) or statistically different from heterozygous mutant ([x]p<0.05, [xx]p<0.01, [xxx]p<0.001). Full genotypes given in legend to *Table 1*.

| Line | Age (days) | Flight index 22°C | Flight index 15°C | WBF 22°C (Hz) | WBF 15°C (Hz) |
|---|---|---|---|---|---|
| *PwMhc2* control | 2 | 4.1 ± 0.16 (111) | 2.3 ± 0.12 (111) | 196 ± 3 (10) | 156 ± 3 (10) |
| *R146N-15/+* heterozygote | 2 | 3.5 ± 0.16 (111)*** | 2.0 ± 0.10 (111) | 194 ± 3 (10) | 150 ± 2 (10) |
| *R146N-15* homozygote | 2 | 1.1 ± 0.11 (111)*** [xxx] | 1.1 ± 0.10 (111)***[xxx] | 182 ± 3 (10)*** [xx] | 141 ± 4 (10)** [x] |

DOI: https://doi.org/10.7554/eLife.38064.014

homozygotes at both temperatures, with a statistically significant 15% reduction compared to the control line at 22°C and no significant difference with the control line at 15°C (*Table 4*). The flight ability of the other two lines of homozygous R146N mutants also was severely diminished (*Supplementary file 2*). Impairment for all lines increased with age in a statistically significant manner, as flight index at 7 days decreased to about half that at 2 days (*Supplementary file 2*). In contrast, the flight index of control organisms showed a significant decrease of only 11% during the same timeframe (*Supplementary file 2*).

We also assessed wing beat frequency (WBF) at 2 days of age and found a significant reduction in the homozygous *R146N* mutants, with 10% (15°C) and 7% (22°C) decreases compared to controls (*Table 4*). The *R146N/+* heterozygous mutant showed no significant decrease in WBF at either temperature. However, there was a significant difference between the WBFs of the heterozygous and homozygous mutants at both temperatures. Thus, R146N myosin detrimentally affects WBF and flight ability. Homozygotes display more severe phenotypes than heterozygotes.

## Structural changes in *R146N/+* heart tubes

We next assessed the structure of adult hearts of *R146N/+* heterozygous flies by transmission electron microcopy to determine the effects of the dominant myosin mutation on cardiac ultrastructure. Two distinct layers are observed in transverse sections taken between the second and third abdominal segments (*Figure 6*). The lumen-facing layer is composed of contractile cardiomyocytes that contain a circumferential array of myofibrils oriented perpendicularly to the anterior-posterior axis of the heart. A layer of structurally supportive ventral-longitudinal skeletal muscle is located ventral to the cardiomyocyte layer (*Figure 6*, VL). In heterozygous controls (*PwMhc2/+*), both young 1-week-old and aged 3-week-old flies showed intact myofibrils with characteristic discontinuous Z-lines (*Achal et al., 2016*) (*Figure 6* left). In contrast, *R146N-15/+* heterozygotes displayed areas of

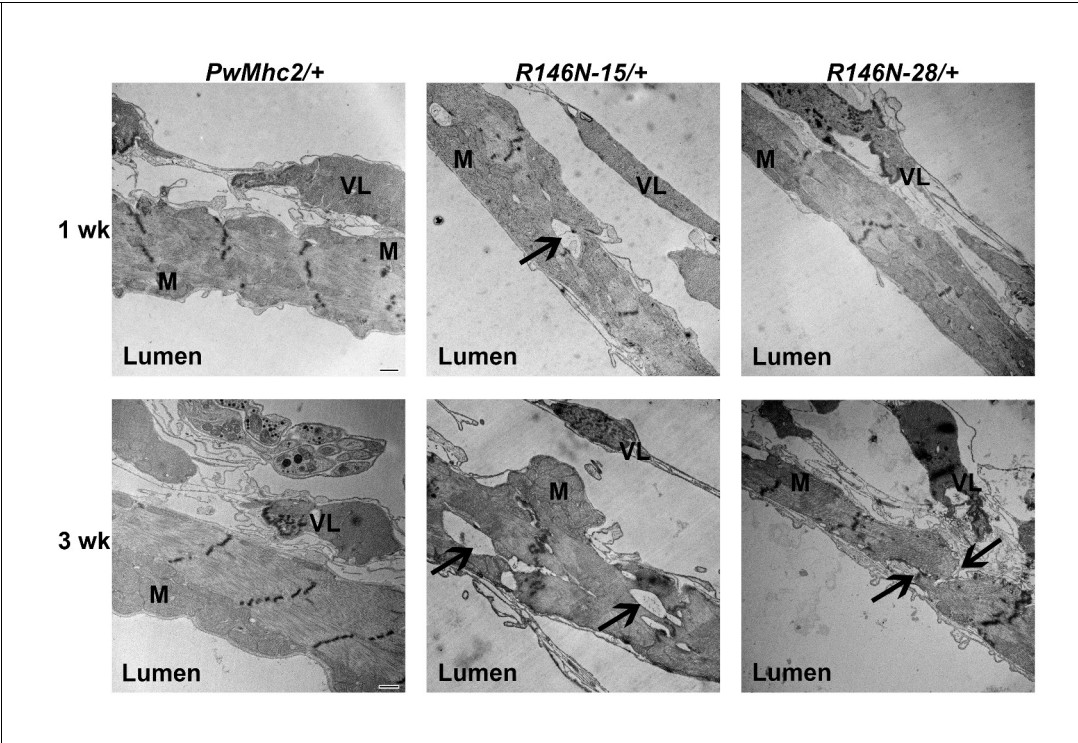

**Figure 6.** Transmission electron microscopy images of hearts of 1- and 3-week-old *PwMhc2/+* control and *R146N/+* mutant heterozygote lines. Micrographs show transverse images through the cardiac tube between the second and third sets of ostia. M, mitochondria. VL, supportive ventral-longitudinal fibers. 'Lumen' refers to the hemolymph-containing center of the heart tube. Arrows indicate areas of myofibrillar discontinuities, which are only observed in mutant hearts. Scale bars, 500 nm. Full genotypes are shown in parentheses: *PwMhc2/+* (P{PwMhc2}; *Mhc¹*/+); *R146N/+* (*Mhc¹*/+; P{R146N}).
DOI: https://doi.org/10.7554/eLife.38064.015

myofibrillar discontinuity, where filamentous fields appear to have been pulled apart (*Figure 6* center, arrows) and these defects worsened with aging. Although *R146N-28/+* hearts showed essentially normal myofibrillar integrity and organization in young flies, they displayed myofibrillar discontinuities in aged flies (*Figure 6* right, arrows). This suggests that defects in older mutant flies are not due to gross assembly defects but occur during the aging process. To assess whether concentric cardiac hypertrophy (addition of myofibrils in parallel) occurs in *R146N/+* heterozygotes, we measured the average cardiomyocyte thickness in ventral and dorsal areas and compared values between young and aged flies for mutant lines and controls. Cardiomyocyte thickness in both young and aged mutant heterozygotes showed no statistically significant differences compared to controls (*Supplementary file 3*). Further, we observed no evidence of eccentric hypertrophy (addition of sarcomeres in series), given that our cardiac physiological analysis showed no increases in chamber dimensions in young or old mutant hearts under basal conditions (*Figure 7A–C*), and a dilated phenotype was not observed after complete relaxation with EGTA+ blebbistatin (*Figure 8A–C*). Hence the R146N myosin mutation caused progressive defects in myofibrillar continuity, but did not yield hypertrophy in the *Drosophila* heart.

## The R146N mutation causes cardiac restriction and diastolic dysfunction

To investigate the pathophysiological consequences of the dominant R146N HCM mutation on the adult *Drosophila* heart, we surgically exposed, imaged, and assessed wall movement in semi-intact *R146N/+* heterozygous flies. The effects of mutant myosin expression on cardiac dimensions and contractile performance, at 1- and 3-weeks of age, were quantified and compared to those resulting from wild-type transgenic myosin expression in age-matched *PwMhc2/+* controls (*Figure 7A–C*). The diastolic and systolic diameters across the heart wall of 1-week-old control (*PwMhc2/+*) *Drosophila* were $65.68 \pm 0.73$ and $39.60 \pm 0.46$ μm, respectively, and were $67.58 \pm 0.80$ and $43.49 \pm 0.67$ μm in 3-week-old animals (*Supplementary file 4*). R146N myosin expression triggered a significant reduction in dimensions at both ages. The diastolic and systolic diameters of *R146N-15/*

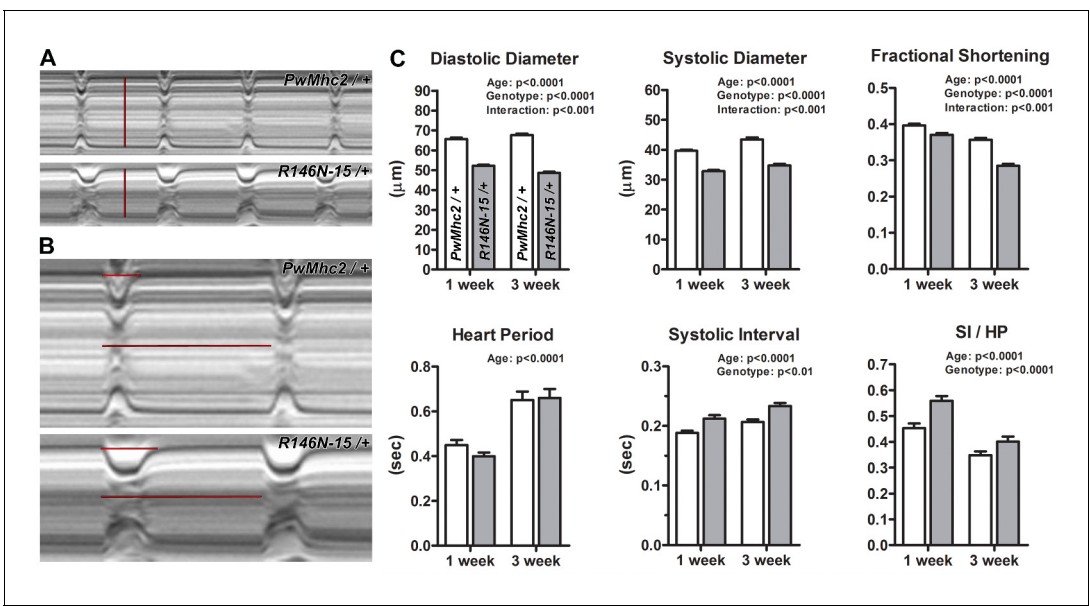

**Figure 7.** Expression of R146N myosin engenders a restricted cardiac physiology with diastolic dysfunction. M-mode kymograms generated from high-speed videos of beating 3-week-old *PwMhc2/+* and *R146N-15/+* hearts. (A) Vertical red lines delineate diastolic diameters. Mutant hearts display cardiac restriction. (B) Horizontal lines demarcate systolic intervals (SI, top) and heart periods (HP, bottom). Mutant hearts display prolonged systolic phases. (C) *R146N-15/+ Drosophila* exhibit highly significant alterations in several cardiac functional parameters relative to *PwMhc2/+* control flies. Both lines were examined at 1- and 3-weeks of age to track potential physiological defects and cardiac remodeling over time. Decreased cardiac dimensions, fractional shortening, and extended periods of systole are observed in 1- and 3-week-old mutants relative to controls. Data are presented as mean ±S.E.M. (N = 40–44 for each genotype and age group) and were evaluated using two-way ANOVAs with Bonferroni multiple comparisons tests. Significance was assessed at p<0.05. Full genotypes are shown in parentheses: *PwMhc2/+* (P{PwMhc2}; Mhc[1]/+); *R146N/+* (Mhc[1]/+; P{R146N}).
DOI: https://doi.org/10.7554/eLife.38064.016

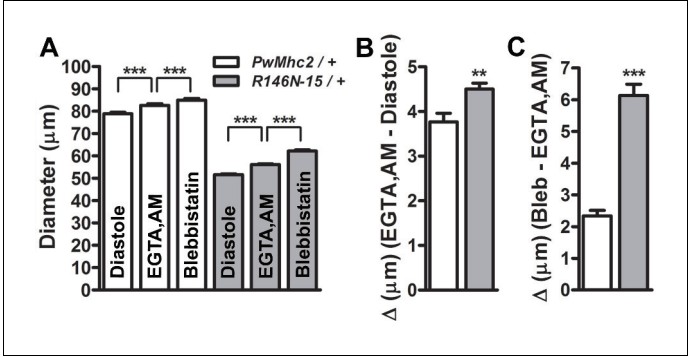

**Figure 8.** Excessive Ca$^{2+}$-dependent and Ca$^{2+}$-independent actomyosin associations during diastole promote incomplete relaxation of *R146N-15/+* cardiomyocytes. (A) Significant, incremental increases in cardiac diameters (repeated measures ANOVA with Bonferroni's post-hoc tests) occur for *PwMhc2/+* and *R146N-15/+* hearts following extra- and intra-cellular Ca$^{2+}$ chelation (EGTA/EGTA,AM) and again, upon blebbistatin incubation (\*\*\*p≤0.0001). (B) The change in diameter across the heart wall in response to EGTA/EGTA,AM is higher in *R146N-15/+* cardiac tubes, indicative of impaired Ca$^{2+}$ homeostasis and a Ca$^{2+}$-dependent increase in resting myocardial tension contributing to the restricted cardiac diameter (\*\*p≤0.01). (C) Addition of blebbistatin after chelation of extra- and intracellular Ca$^{2+}$ also prompted a significantly greater increase in cardiac dimensions of *R146N-15/+* hearts relative to that observed for *PwMhc2/+* hearts (\*\*\*p≤0.0001). This implies that excessive Ca$^{2+}$-independent cross-bridge cycling occurs in *R146N-15/+* hearts, resulting in further enhanced resting basal tension and incomplete relaxation. Data are presented as mean ± S.E.M. (*N* = 20). The effects of EGTA, EGTA-AM and blebbistatin treatment on diameters were evaluated using paired t-tests. Unpaired two-tailed t-tests were used to distinguish significant differences in cardiac responses to EGTA, EGTA.AM and blebbistatin between *PwMhc2/+* and *R146N-15/+* hearts. Significance was assessed at p<0.05. Full genotypes are shown in parentheses: *PwMhc2/+* (P{PwMhc2}; Mhc$^1$/+); *R146N/+* (Mhc$^1$/+; P{R146N}).
DOI: https://doi.org/10.7554/eLife.38064.017

+at 1 week were 52.13 ± 0.53 and 32.83 ± 0.45 µm, respectively, and were 48.62 ± 0.64 and 34.76 ± 0.53 µm at 3 weeks. The greater mutational effect on diastolic vs. systolic diameter resulted in 7.5% and 22% decreases in fractional shortening at 1 and 3 weeks of age relative to controls (*Figure 7C*). Similar results were found for *R146N-28/+* flies (*Supplementary file 4*). The age-related decrease in fractional shortening suggests progressive remodeling in the mutant, which correlates with the decreasing myofibrillar continuity we observed.

The heart period, which is the time required for a complete cardiac cycle consisting of a diastolic and subsequent systolic phase, was not significantly different between mutant and control flies at either age tested (*Figure 7C*). The systolic interval, however, was 0.19 ± 0.01 and 0.21 ± 0.01 s for 1- and 3-week-old control flies and 0.21 ± 0.01 and 0.23 ± 0.01 s for *R146N-15/+* heterozygotes. Therefore, the proportion of time during the cardiac cycle that was spent generating tension, calculated by determining the ratio of systolic interval to heart period (SI/HP), was significantly greater for the mutant heart tubes relative to controls at all ages studied (*Figure 7C*). Together, these findings illustrate that the *R146N/+* hearts exhibit reduced cardiac diameters and potentially elevated tone during diastole, as well as prolonged periods of systolic tension generation, which are indicative of restrictive physiology, diastolic dysfunction, and impaired relaxation (*Cammarato et al., 2008*; *Viswanathan et al., 2014*; *Viswanathan et al., 2017*).

## The R146N mutation reduces cardiac diameters during diastole by calcium-dependent and calcium-independent mechanisms

To further investigate the molecular basis of diastolic restriction and potentially altered resting tone in the *R146N/+* heterozygote hearts, we compared the responses of three-week-old control and mutant cardiac tubes to treatments with EGTA-AM, a cell permeable chelator of intracellular Ca$^{2+}$ (*Johnson et al., 1997*) and blebbistatin, a small molecule myosin inhibitor (*Fedorov et al., 2007*; *Limouze et al., 2004*). Upon incubation with artificial hemolymph supplemented with 10 mM EGTA and 100 µM EGTA-AM, which removes extra- and intracellular Ca$^{2+}$, the hearts ceased beating and 'relaxed', as manifest by an increase in diameter across the wall of both control and *R146N/+* hearts

relative to diastole (*Figure 8A*). Interestingly, there was a significantly amplified relaxation response in *R146N/+*hearts (Δ = 4.50 ± 0.13 μm) relative to controls (Δ = 3.77 ± 0.19 μm) (*Figure 8B*).

We previously demonstrated in vivo, that a small population of residual cross-bridges actively cycles and generates force, and helps establish basal mechanical tone in wild-type *Drosophila* myocardium during diastole (*Viswanathan et al., 2014*; *Viswanathan et al., 2017*). In this regard, subsequent treatment of control and mutant hearts with artificial hemolymph containing 10 mM EGTA, 100 μM EGTA-AM, and 100 μM blebbistatin generated a second 'relaxation' response (*Figure 8A*). Comparing the cardiac diameters after extra- and intra-cellular Ca$^{2+}$ chelation and following blebbistatin treatment revealed a significant increase of roughly 2.8% (Δ = 2.34 ± 0.17 μm) in control hearts (*Figure 8C*). Notably, there was a strikingly augmented response to blebbistatin treatment in the *R146N/+* heterozygote hearts that led to a ~ 10% (Δ = 6.14 ± 0.35 μm) increase in heart tube diameter (*Figure 8C*). Comparing the change in diameters between the two genotypes revealed that the response to blebbistatin is significantly greater in the *R146N/+* mutant hearts vs. that for controls (*Figure 8C*). Our observations suggest that, in addition to elevated diastolic Ca$^{2+}$ and Ca$^{2+}$-activated cross-bridge cycling, excessive Ca$^{2+}$-independent cross-bridges contribute to diastolic dysfunction and inadequate relaxation in the *R146N/+* mutant hearts.

Overall, we utilized an integrative approach in the *Drosophila* system to determine the effects of a myosin HCM mutation on myosin function, skeletal muscle structure, muscle mechanics and cardiac physiology. We found that altering modeled interactions between the mutant R146N residue and the lever arm leads to increased basal ATPase activity and changes to myosin cross-bridge kinetics that disrupt actin motility, myofibril power output, myofibril stability, flight ability and cardiac structure and function. Our analyses provide insight into how a myosin mutation translates into mutant phenotypes, thereby elucidating underlying mechanisms of HCM.

## Discussion

We leveraged the strengths of the *Drosophila* myosin system to determine new phenotypic and mechanistic insights into HCM, a common disease of young adults. The *Drosophila* model is an outstanding tool for integrative analysis of contractile protein mutations and for dissecting the mechanistic basis of disease, allowing an understanding from the molecular through the whole organism level. In our HCM model, the *Drosophila* heart tube shows a restricted phenotype, which, like abnormalities in the mutated IFM, develops from prolonged cross-bridge binding to actin. The restricted cardiac tube and diastolic dysfunction in this model of human HCM is congruent with phenotypes observed when expressing myosin with enhanced function (*Cammarato et al., 2008*) or tropomyosin that is inefficient at blocking the myosin binding site on actin (*Viswanathan et al., 2014*). This suggests that *Drosophila* is an excellent analytical system for screening mutations to assess their potential for causing human HCM, which are typically expected to enhance contractility (*Adhikari et al., 2016*; *Nag et al., 2017*; *Spudich, 2014*).

Integrating the results from our ATPase, actin motility and muscle mechanical studies reveals that there are alterations to the myosin cross-bridge cycle caused by the R146N mutation. Muscle mechanics measurements showed an increase in apparent rate constant 2πb, which, according to the modeling of *Kawai and Brandt (1980)* is influenced by rates associated with myosin attachment to actin, including binding rate, phosphate release and the power stroke (*Figure 5—figure supplement 1*). The enhanced basal myosin ATPase rates suggest that phosphate release rate has been increased, since this parameter is rate limiting for basal ATPase rate. This would yield an increased speed of weak to strong actin binding.

Our modeling of molecular interactions within the myosin molecule suggests a destabilization of the pre-power stroke state, which could account for the increased affinity of the R146N myosin ADP. Pi state for actin and faster phosphate release rate. *Drosophila* myosin residue R146 normally has charge-based interactions with E774 and E775 of the myosin lever arm at the pre-power stroke state (*Figure 1C*). This interaction is abolished by the R146N mutation (*Figure 1D*), which would reduce the stability of the pre-power stroke molecule. This could drive the cycle forward by enhancing ATP hydrolysis and the production of cross-bridges, possibly contributing to prolonged actin binding. A similar mechanism may occur for human β-cardiac myosin as well, since the analogous K146 residue is also predicted to interact with E775 in the pre-power stroke state (*Figure 1—figure supplement 1A*) and this contact is destroyed by the K146N mutation (*Figure 1—figure supplement 1B*).

Interestingly, analysis of another HCM mutation suggests a similar mode of action as for R146N (*Guhathakurta et al., 2017*). Time-resolved fluorescence resonance energy transfer analysis of myosin containing the E56G mutation in the lever-arm-binding essential light chain showed a reduction in pre-power stroke state molecules and enhanced levels of those in the post-power stroke state in the presence of ATP. As with R146N, this yielded greater actin attachment and an increased duty ratio (*Guhathakurta et al., 2017*).

Recent studies of omecamtiv mecarbil, a compound that enhances cardiac myosin power output (*Liu et al., 2015*; *Planelles-Herrero et al., 2017*; *Shen et al., 2010*), are also relevant to understanding the molecular basis of the R146N mutant phenotypes. As shown in *Figure 1—figure supplement 1C*, the compound binds in a pocket at the nexus of the N-terminus (including K146), the relay helix and the converter domain/lever arm (including residue E774) (*Planelles-Herrero et al., 2017*). Omecamtiv mecarbil enhances the actin attachment rate and the duty ratio and slows in vitro motility (*Liu et al., 2015*), similar to the effects that we observed with R146N myosin. In contrast to our findings, ATPase activity in the absence of actin is inhibited by the compound (*Liu et al., 2015*). However, the rate of phosphate release is increased upon actin binding. While the R146N mutation enhances basal activity, which may arise from destabilization of the pre-power stroke state, omecamtiv mecarbil appears to stabilize this state prior to actin interaction, including facilitation of a weak charge-based interaction between K146 and E774 (*Figure 1—figure supplement 1C*). Further clinical evidence supports the importance of these interactions in relationship to HCM, in that mutation of E774 to V774 also yields HCM (*Moric et al., 2003*). More rapid cross-bridge formation due to the R146N or E774V mutations destabilizing the pre-power stroke state coupled with decreased detachment rates (see below) stabilize actin-myosin interaction, leading to enhanced force production and decreased actin filament in vitro motility. A similar outcome could arise from omecamtiv mecarbil increasing the number of molecules in the pre-power stroke state, which would increase the number of attached cross-bridges as the myosin molecules move through the mechanochemical cycle.

Although myosin attachment (*Brizendine et al., 2017*) and detachment rates (*Harris and Warshaw, 1993*) both influence the duty ratio (the fraction of time myosin is strongly bound to actin during a mechanochemical cycle), the latter is thought to have the major influence on in vitro motility at high myosin concentrations. Thus, the ~50% decrease in the rate of actin movement in vitro for R146N myosin (*Figure 2F*) is likely facilitated by the decreased actin detachment rate suggested by our fiber mechanics measurements. We observed a decrease in apparent rate constant 2πc (*Table 3*), a measure of rates associated with myosin detachment from actin, which includes detachment induced by ADP release and ATP binding (*Kawai and Brandt, 1980*) (*Figure 5—figure supplement 1*). Further, our observed increase in ATP $K_m$ in the fiber studies (*Figure 5A*) suggests decreased ATP affinity, which would slow the ATP-induced detachment rate of myosin from actin. This is supported by the trends showing decreases in $f_{max}$ with increasing phosphate concentration for the mutant (*Figure 5B*), which does not occur in control myosin. $f_{max}$ is most likely being slowed by phosphate competing with ATP for the rigor state (*Pate and Cooke, 1989*). Normally, increasing phosphate either has no effect (as seen in the control) or causes $f_{max}$ and 2πb to increase (as seen in many slow vertebrate muscle types [*Galler et al., 2005*; *Kawai et al., 1993*; *Siemankowski et al., 1985*]). Overall, the increase in attachment and decrease in detachment rate, in concert with no change in actin-activated ATPase rate arising from the mutation, indicate an increase in duty ratio. In addition to slowed in vitro motility, an increase in duty ratio would cause the increased fiber tension generation and stiffness that we observed from analysis of elastic modulus values (*Figure 5C,D*).

The increased duty ratio is also expected to negatively influence mutant IFM power generation (*Figure 4A*). The major cause of power loss is likely decreased net work production, given that there was relatively little change in optimal frequency for muscle power generation ($f_{max}$). Lack of change in $f_{max}$ may be due to increased actin attachment and phosphate release being balanced by reduced ATP-induced detachment rate. Work production, however, was decreased proportionally to power for homozygous fibers examined via sinusoidal analysis, and for both homozygous and heterozygous fibers when measured using the work loop technique (*Figure 4B*, *Table 2*). When amplitude and length change conditions were optimized for power generation, the ratio of work absorption to work generation was increased in the mutants compared to control fibers. Tellingly, decreasing optimal muscle length change amplitude allowed the mutant fibers to generate more power (compare optimal power work loops with work loops performed under the identical length change and frequency conditions). This indicates that less muscle stretching was beneficial because it decreased

work absorbed, resulting in more net work and hence more power. It appears that increased time of cross-bridge attachment causes cross-bridge elements to act more like brakes or shock absorbers during cyclical power generation. These abnormally high stresses on the myofilaments likely result in the disruption of sarcomere integrity that develops over time in the IFM expressing the mutant myosin (*Figure 3*), as well as the reduced flight ability and WBF (*Table 4*).

Our analysis of the heart tube in *R146N/+* heterozygotes produced results that support the thesis that an increased myosin duty ratio yields significant mutant phenotypes. While the hearts in the mutants did not show a change in heart period (suggesting no overall change in kinetics, in agreement with no change for $f_{max}$ in IFM and for actin-activated ATPase), the heart spent a greater portion of each heart period in systole compared to the control (*Figure 7*). This is consistent with prolonged contractions arising from increased myosin attachment kinetics and a decreased detachment rate. An enhanced attachment rate would also increase the rate of force generation, which may partially account for the restricted cardiac phenotype. The reduced fractional shortening observed for the *R146N/+* heart may correspond to its optimal pumping mode, mimicking the improved IFM fiber mechanical performance at shorter muscle length changes. At the ultrastructural level, the observed myofibrillar discontinuity in cardiomyocytes (*Figure 6*) could be caused by the same mechanism as IFM degeneration, that is, excessive tension generation disrupting the integrity of the sarcomeric ultrastructure. We did not detect hypertrophy of the cardiomyocytes in *R146N/ +* heterozygotes, although it has been shown that hypertrophy can occur in the conical chamber of the *Drosophila* heart as a result of abnormal receptor tyrosine kinase pathway signaling (*Yu et al., 2013*). It is important to note, however, that there is significant variability in the presence and degree of hypertrophy detected in patients carrying the *146N* mutation (personal communication from Jodie Ingles, Sydney Medical School), suggesting that interacting genes or environment may play a role in hypertrophy.

Interestingly, our data suggest that cross-bridges in the *Drosophila* heart bind to thin filaments via $Ca^{2+}$-dependent and $Ca^{2+}$-independent mechanisms during diastole. This contributes to impaired relaxation. Our results imply that small amounts of diastolic $Ca^{2+}$ promote contraction and yield slightly shortened cardiomyocytes at rest in both mutant and control lines. However, based upon the greater diameter increase (greater tension drop) in *R146N/+* hearts compared to control hearts upon extra- and intracellular $Ca^{2+}$ chelation with EGTA/EGTA,AM (*Figure 8B*), disruption of $Ca^{2+}$-handling in the mutant appears likely, which may promote the restricted cardiac tube phenotype. Therefore, it is possible that perturbation of $Ca^{2+}$-handling in *R146N/+* mutant fly hearts, as is frequently observed in human cardiomyopathies (*Kranias and Bers, 2007*), results in excessive diastolic $Ca^{2+}$ levels, yielding enhancement of cell shortening and elevated basal tone. Additionally, the higher duty ratio of R146N myosin could indirectly enhance the $Ca^{2+}$ sensitivity of the cross-bridges through thin filament activation. Full thin filament activation requires both $Ca^{2+}$ binding to troponin and strong binding of cross-bridges. Thus, a higher duty-ratio myosin will have a greater number of myosin molecules strongly bound at a lower $Ca^{2+}$ concentration than a lower-duty-ratio myosin. We experimentally demonstrated this when we expressed the high-duty-ratio EMB myosin in the jump muscle, which caused enhanced $Ca^{2+}$ sensitivity for tension development compared to jump muscle myosin (*Eldred et al., 2010*). Thus, the same concentration of $Ca^{2+}$ in the *R146N/+* mutant heart would generate higher force than in the control, yielding decreased cardiac diameters.

Our analysis with blebbistatin, which impedes acto-myosin cycling (*Kovács et al., 2004*), indicates that a portion of the cross-bridges responsible for diastolic tone is calcium-independent in both control and *R146N/+* hearts. Treatment with the drug (*Figure 8C*) shows that there is a greater diameter increase in *R146N/+* hearts than in control hearts, suggesting an enhanced number of $Ca^{2+}$-independent cross-bridges. This enhancement likely arises from the higher on-rate (actin affinity), which contributes to the restricted phenotype and impaired relaxation observed in *R146N/+* mutant hearts.

Increased cross-bridge availability could also result from disruption of the super-relaxed state that has been documented in both skeletal and cardiac muscles (*Hooijman et al., 2011*; *Stewart et al., 2010*). This state correlates with the presence of the interacting head motif, wherein the dimeric heads of myosin molecules interact to reduce their actin binding and ATPase activity (*Trivedi et al., 2018*; *Woodhead et al., 2005*). Disruption of this motif by mutation is linked to myosin activation that correlates with HCM (*Alamo et al., 2017b*; *Nag et al., 2017*). Interestingly, myosin is found in its pre-power stroke configuration in the interacting head motif (*Alamo et al., 2017a*; *Alamo et al., 2016*) and disturbance of this state of filament packing by the R146N mutation could reduce the

number of molecules in the super-relaxed state. This would enhance myosin activity within the myofibril, leading to the phenotypes we observed. *Drosophila* myosins are indeed capable of entering the interacting head motif (*Lee et al., 2018*), but the presence of the super-relaxed state in thick filaments or muscle fibers is yet to be documented.

In summary, our integrative analysis of the *Drosophila* analogue of the K146N human HCM mutation revealed increases in several, but not all, contractile parameters, as posited for a general mechanism of action for HCM mutations in humans (*Spudich, 2014*). Notably, enhancements in basal ATPase rate, length of systole and cross-bridge binding are concordant with these expectations. Further, the restricted phenotype and reduced fractional shortening observed in the *Drosophila* heart mimic the diastolic dysfunction/impaired relaxation and reduced ejection fraction that can accompany HCM (*Davis et al., 2016*; *Maron, 2002*; *Maron and Maron, 2013*; *Masarone et al., 2018*). The restricted phenotype observed in our model of an HCM mutant, as well as the restricted phenotypes arising from a hyperactive myosin (*Cammarato et al., 2008*) or mutant troponin T or actin that enhance myosin binding (*Viswanathan et al., 2014*; *Viswanathan et al., 2017*), suggest that restriction is a common phenotype produced in the *Drosophila* cardiac tube due to over-active cross-bridges. If all or most HCM mutations have increased force production as their basis, the restricted cardiac tube phenotype in *Drosophila* could serve as a useful screening tool for determining whether specific human mutations are causative of HCM. The fly might also serve to illuminate conserved signaling pathways that lead from contractile protein mutations to HCM. Further, the amelioration of the mutant phenotype through genetic or drug screens in *Drosophila* may yield insights into potential treatments for human HCM.

# Materials and methods

## Protein structure modeling

Scallop muscle myosin II crystal structures in the pre-power stroke state (PDB 1QVI) (*Gourinath et al., 2003*) and the actin-detached post-power stroke state (PDB 1KK8) were used as templates to predict *Drosophila* and human myosin structures (*Himmel et al., 2002*). Myosin S1 amino acid sequences were modeled using the SWISS-MODEL homology modeling server (http://swissmodel.expasy.org/) (*Arnold et al., 2006*). Structures were viewed using PyMOL (http://www.pymol.org; DeLano Scientific, Palo Alto, CA).

## DNA constructs

A *P* element-containing *Mhc* transgene with the R146N mutation was constructed using standard cloning techniques along with site directed mutagenesis. The wild-type genomic construct PwMhc2 (*Swank et al., 2000*) was digested with Eag I to produced two subclones, PwMhc-5' and pMhc3'. The PwMhc-5' subclone contains an 11.3 kb Eag I *Mhc* fragment cloned into the pCaSpeR vector (*Thummel and Pirrotta, 1992*). The PMhc-3' subclone contains a 12.5 kb Eag I *Mhc* fragment cloned into pBluescriptKS Eag I site (Stratagene, La Jolla, CA). The PwMhc-5' subclone was further digested with Xho I and Avr II. A 6.8 kb Xho I-Avr II digested fragment from PwMhc-5' was gel isolated and ligated into an Xho I-Avr II site in pLitmus 28I vector (New England Biolabs, Ipswich, MA) to produce pMhc-5'-XA. The PwMhc-5'-XA subclone was further digested with Pst I and Avr II. A 4.3 kb Pst I-Avr II digested fragment from PwMhc-5'-XA was gel isolated and ligated into an Pst I-Avr II site in pLitmus 28I vector, to produce pMhc-5'-PA. The PwMhc-5'-PA subclone was further digested with Pst I and Age I. A 1.6 kb Pst I-Age I digested fragment from PwMhc-5'-PA was gel isolated and ligated into an Pst I-Age I site in pLitmus 28I vector, to produce pMhc-5'-PA1.6. The pMhc-5'-PA1.6 subclone was subjected to site-directed mutagenesis using the QuickChange II kit (Stratagene) and exon specific-primer 5'-CCGTGGCAAG**AAC**CGTAATGAGG-3' containing the R146N nucleotide coding change (bold) to yield, pR146N-5'PA-1.6. Upon sequence confirmation of the R146N site-directed mutagenesis product, the pR146N-5'-PA-1.6 subclone was digested with Pst I and Age I. The 1.6 kb R146N fragment from this digest was used to replace the wild-type Pst I-Age I fragment of pMhc-5'PA. The resulting clone was digested with Pst I-Avr II and the Pst I-Avr II fragment was used to replace the wild-type Pst I-Avr II fragment of pMhc-5'-XA. The resulting clone was digested with Xho I-Avr II and the isolated Xho I-Avr II fragment was used to replace the wild-type Xho I-Avr II fragment of PwMhc-5'. The resulting clone, PwMhcR146N-5', was digested with Eag I. The wild-type

3' end subclone pMhc-3' also was digested with Eag I. The 12.5 kb Eag I fragment was gel isolated and ligated into the Eag I of PwMhcR146N-5', to yield PwMhcR146N. The entire coding region and all ligation sites of the final PwMhcR146N plasmid were confirmed by DNA sequencing (Eton Bioscience, San Diego, CA) prior to *P* element transformation.

## *P* element transformation of *Mhc* genes

Transgenic lines for PwMhcR146N were generated by *P* element-mediated germline transformation (*Rubin and Spradling, 1982*) by BestGene, Inc. (Chino Hills, CA). BestGene injected 1200 embryos with the PwMhcR146N transgene, and 29 transgenic lines were obtained. Each insert was mapped to its chromosomal location using balancer chromosomes and standard genetic crosses. Ten inserts mapped to the second chromosome, 3 to the X chromosome and 16 to the third chromosome. Three independent lines that mapped to the third chromosome were crossed into *Mhc*[10] background, which is null for myosin heavy chain in IFM and TDT muscle due to mutation of the endogenous *Mhc* gene, which is located on the second chromosome (*Collier et al., 1990*).

## Reverse transcription polymerase chain reaction

RT-PCR was employed to verify that *Mhc* transcripts from the *PwMhcR146N* transgenic lines contained the appropriate site-directed nucleotide changes in exon four and that flanking alternative exons 3 and 7 were spliced correctly. RNA was prepared by $LiCl_2$ extraction (*Becker et al., 1992*) from 2-day-old adult female transgenic flies. cDNAs were generated for each line using the Protoscript cDNA synthesis kit (New England Biolabs). A reverse primer specific to exon 8 (5'-GTTCG TCACCCAGGGCCGTA-3') and a forward primer specific to exon 2 (5'-TGGATCCCCGACGA-GAAGGA-3') were used to generate cDNA. Briefly, 3 μmol of reverse specific primer were mixed with 0.5 μg of total RNA from each transgenic line. PCR was performed using 1 μl of cDNA and 3 μmol of forward and reverse primers under the following conditions: 60 s at 94° C then 30 cycles of: 30 s at 94° C, 30 s at 55° C and 2 min at 68° C. RT-PCR products were sequenced by Eton Bioscience.

## Myosin expression levels

To determine myosin protein expression levels for each transgene in an *Mhc*[10] background, SDS polyacrylamide gel electrophoresis was utilized as previously described (*O'Donnell et al., 1989*). Upper thoraces from six 2-day-old homozygous female flies were homogenized in 60 μl SDS gel buffer. Six μl of sample were loaded on a 9% polyacrylamide gel. This was repeated five different times each time using a freshly prepared sample. Protein accumulation was determined using Coomassie blue stained gels that were digitally scanned and analyzed on NIH Image J software. Values are expressed relative to the control transgene ±S.E.M.

## Actin and myosin preparation, ATPase activity and in vitro motility

As previously reported, actin was prepared from chicken skeletal muscle after multiple steps of polymerization–depolymerization (*Kronert et al., 2014*; *Kronert et al., 2015*). Myosin was isolated from dissected dorsolongitudinal IFMs of ~250 transgenic flies and its concentration was determined by spectrophotometry (*Kronert et al., 2014*; *Kronert et al., 2015*; *Swank et al., 2001*). A final concentration of 2.0 μg/μl myosin was used for ATPase assays and 0.5 μg/μl was used for in vitro motility assays. Steady-state calcium, magnesium and actin-stimulated magnesium ATPase activities of myosin were obtained using [γ-$^{32}$P]-ATP as previously described (*Kronert et al., 2014*; *Kronert et al., 2015*; *Swank et al., 2001*). Basal Mg-ATPase activities obtained in the absence of actin and actin-stimulated Mg-ATPase was determined using increasing concentrations of filamentous actin. Basal Mg-ATPase values were subtracted from all data points, which were then fit with the Michaelis–Menten equation to determine actin-stimulated ATPase ($V_{max}$) and actin affinity relative to ATPase ($K_m$). Catalytic efficiency (defined as $V_{max}/K_m$) was determined as previously reported (*Kronert et al., 2014*; *Kronert et al., 2015*). In vitro motility assays were carried out by adding ATP and filamentous actin labeled with fluorescent phalloidin to nitrocellulose treated cover slips coated with wild-type or mutant myosin (*Kronert et al., 2014*; *Kronert et al., 2015*; *Swank et al., 2001*). Video sequences captured under fluorescence optics were analyzed computationally to determine actin-sliding velocity. Mean values from five independent experiments (two assays for each) for the mutant and wild-

type ATPase or from three independent motility assays (at least 30 filaments/assay) were compared for statistically significant differences (p<0.05) by Student's t-tests.

## Transmission electron microscopy

To determine the effects of transgene expression on IFM structure in an $Mhc^{10}$ null background, transmission electron microscopy was performed as previously described (*O'Donnell and Bernstein, 1988*). We examined four different stages of development; late stage pupae, 2-hr-old adults, 2-day-old adults and 1-week-old adults. Cross-sections and longitudinal sections were obtained from females for each homozygous transgenic line, with at least three different samples examined for each of the three transgenic lines. One myofibril is shown in each panel and is representative of the entire population at the given stage of development. Images were obtained using a FEI Tecnai 12 transmission electron microscope with a TVIPS 214 high-resolution camera.

For cardiac electron microscopy, heart samples at 1 or 3 weeks of age (three per age per genotype) were surgically exposed in an oxygenated artificial hemolymph solution and fixed using a modified procedure as described previously (*Achal et al., 2016*). Rhythmic beating was observed to confirm structural integrity of samples. Hearts were relaxed in 10 mM EGTA, fixed in primary fixative (3% formaldehyde, 3% glutaraldehyde, 0.1 M cacodylate buffer pH 7.4), washed 6x in 0.1 M cacodylate buffer pH 7.4, then bathed in secondary fixative (1% $OsO_4$, 10 mM $MgCl_2$, 100 mM phosphate buffer pH 7.4) and washed 3x in 0.1 M cacodylate buffer pH 7.4. The samples were dehydrated in an acetone series, oriented and embedded in Epon-filled BEEM capsules, and polymerized at 60°C under vacuum. Thin sections ($\leq$50 nm) were obtained using a Leica ultramicrotome, picked up on Formvar-coated grids, and stained with 4% uranyl acetate for 10 min. Images were obtained at 120 kV on a FEI Tecnai 12 transmission electron microscope.

Transverse sections of the heart were utilized to determine average cardiac thickness between the second and third abdominal segments. For each measurement, a roughly rectangular area of cardiomyocyte tissue (defined as myofibrillar and mitochondrial material) of 10 μm in length within the inner and outer edges of the cardiac tube was highlighted using Adobe Photoshop. The image was imported into ImageJ to calculate the area of the highlighted region. Dividing the resulting cardiomyocyte area by 10 μm yielded the average cardiac thickness for that highlighted section. At least three images each for dorsal areas and for ventral areas per heart section were analyzed. Three or more transverse sections along the anterior-poster axis at least 10 μm apart were assessed in the same way. Cardiac thickness measurements from dorsal-side or ventral-side images were averaged per section and the data were then averaged for each biological replicate. Data from at least three hearts were averaged per line to yield the final values, which are presented as mean ± SEM. A one-way ANOVA with the Bonferroni correction determined statistical significance (p<0.05).

## Dissection of IFMs for mechanics

Newly eclosed female *PwMhcR146N-15* homozygous or heterozygous flies were collected every half hour and allowed to mature for 2 hr. Control flies were taken from the *PwMhc2* stock. Individual IFM fibers were isolated by dissection as previously described by Swank (*Swank, 2012*). Briefly, the head, abdomen and wings were removed and the thorax was placed into dissection solution (*Swank, 2012*). The thorax was split in half, and the IFM fiber bundle was removed. The fibers were skinned in the dissection solution for 1 hr at 6°C before being transferred to storage solution (same as the dissection solution but without Triton X-100). Individual fibers were then separated from the bundle and split in half with a long, thinly etched tungsten probe. The split fiber was then fitted with a pair of aluminum foil t-clips.

## Muscle mechanics

The clipped fiber was mounted in relaxing solution (pCa 8.0, 12 mM MgATP, 30 mM creatine phosphate, 600 U/ml creatine phosphokinase, 1 mM free $Mg^{2+}$, 5 mM EGTA, 20 mM pH 7.0 BES, 200 mM ionic strength, adjusted with Na methane sulfonate, 1 mM DTT) onto a mechanics apparatus by settling it onto two hooks attached to a motor arm and a force transducer. The muscle was lengthened to be just taught, then stretched 5% beyond this length. The resulting length, width and height were measured to obtain cross-sectional area. Passive tension ($P_o$) was recorded before adding activating solution (the same as relaxing solution except that pCa was adjusted to 5.0). The

maximum power of the muscle was determined by performing sinusoidal analysis (see below), under low amplitude conditions (0.125% muscle length). After every run, the muscle was stretched by 2% muscle length (ML) until the power did not increase by more than 3%. The active tension ($A_o$) at the optimal muscle length was measured and the net tension ($F_o$) was calculated by $A_o-P_o$.

### Work loops

To measure the power under longer ML conditions, the fiber was subjected to a series of ten identical length changes (cycles) over a range of amplitudes (0.25, 0.5, 0.75, 1.0, or 1.25% ML) and frequencies (50, 75, 100, 125, 150, 175, or 200 Hz) to find the optimal conditions for maximum power generation. The power measurement from cycles 7 or 8 was used because the power values become consistent after cycle 6 (*Ramanath et al., 2011*). The fiber was then stretched by 1% ML and sinusoidal analysis repeated at 0.125% amplitude before any further experiments were performed to ensure fiber integrity.

### ATP and phosphate response

The standard fiber activating solution was first exchanged to increase the ATP concentration from 12 mM to 20 mM. Sinusoidal analysis was then performed at decreasing ATP concentrations of 20.0, 15.0, 12.5, 10.0, 7.5, 5.0, 2.5, 2.0, 1.0, 0.75 and 0.5 mM by exchanging with the appropriate amount of 0 mM ATP activating solution. Similar to the ATP response, activating solution was exchanged so that [Pi] = 0 mM, and then the concentration of phosphate was sequentially increased to 1, 2, 4, 8, 12, 16 mM by exchanging in appropriate amounts of a 20 mM phosphate activating solution before performing sinusoidal analysis. All solutions were made so that the ionic strength did not change when calcium, ATP or Pi was varied (*Swank, 2012*). High ATP solution was the same as relaxing solution with the following modifications: pCa 5.0, 20 mM ATP, 37 mM creatine phosphate, 450 U/ml CPK, and 260 mM ionic strength. Low ATP was the same as high ATP, except for 0 mM ATP. High Pi was the same as high ATP, except for 13 mM ATP and 20 mM Pi. Low Pi was the same as high ATP, except 13 mM ATP and 0 mM Pi.

### Sinusoidal analysis

Sinusoidal analysis was performed as previously described (*Swank, 2012*). Briefly, the muscle fiber was oscillated at 50 different frequencies, from 0.5 Hz to 650 Hz. Each oscillation was 0.125% ML and the elastic and viscous modulus were calculated from the amplitude and phase differences. Work and power were calculated as described in *Wang et al. (2014)*. To determine fiber apparent rate constants, the complex modulus was fit with the equation $Y(f) = A (2\pi if/\alpha)k - B if /(b + if) + C if/ (c + if)$ developed by *Kawai and Brandt (1980)*, and described in *Swank (2012)*. The b frequency value is most influenced by work producing steps of the cross-bridge cycle, while the c frequency value is most influenced by work absorbing steps. Frequencies b and c are multiplied by $2\pi$ to convert from frequency to time ($s^{-1}$).

## Flight assays

Two- and 7-day-old homozygotes for each of the three transgenic lines and controls were tested for flight ability in an $Mhc^{10}$ background at 22°C. Flight was determined by the ability to fly up (U), horizontal (H), down (D) or not at all (N) (*Drummond et al., 1991*). As previously described (*Tohtong et al., 1995*) flight index is defined as: 6 U/T+ 4 H/T+ 2 D/T+ 0 N/T where T is the total number of flies tested. Flight abilities and wing beat frequencies of two-day-old *PwMhcR146N-15* homozygotes and heterozygotes were examined at 15°C and 22°C. An optical tachometer was used to measure the wing beat frequency of tethered flies (*Tohtong et al., 1995*). All values represent mean ± SEM. Significance was assessed by a Student's t-test at p<0.05.

## Cardiac analysis of semi-intact *Drosophila* preparations

Cardiac tubes of 1- and 3-week-old female adult flies were surgically exposed under oxygenated artificial hemolymph as described by *Vogler and Ocorr (2009)*. High-speed videos of contracting hearts were taken at ~120 frames per second using a Hamamatsu Orca Flash 2.8 CMOS camera on a Leica DM5000B TL microscope with a 10x (0.30 NA) immersion lens. Physiological parameters were assessed from individual videos using the semiautomatic optical heartbeat analysis (SOHA v.3)

program as previously outlined (*Cammarato et al., 2015*; *Fink et al., 2009*; *Viswanathan et al., 2017*). M-mode kymograms were generated to provide an edge trace documenting the movement of heart wall in the y-axis over time in the x-axis. Values represent mean ± S.E.M. Two-way ANOVAs were employed to test if the effects of genotype and age (or if an interaction exists) were significant for each cardiac parameter investigated. Significance was assessed at $p < 0.05$.

## Quantitation of EGTA.AM and Blebbistatin-induced changes in cardiac dimensions

Beating hearts from 3-week-old heterozygous female *PwMhc2/+* and *PwMhcR146N-15/+* flies were imaged as described above using a 20x (0.50 NA) immersion objective lens and analyzed as previously detailed (*Viswanathan et al., 2014*; *Viswanathan et al., 2017*). The effects of EGTA/EGTA-AM and blebbistatin treatment on cardiac diameters were evaluated using paired t-tests of the matched groups. Unpaired t-tests were used to distinguish significant differences in the cardiac responses to EGTA, EGTA.AM and blebbistatin between *PwMhc2* and mutant MHC expressing hearts. Significance was assessed at $p < 0.05$.

---

# Additional information

## Funding

| Funder | Grant reference number | Author |
| --- | --- | --- |
| National Institutes of Health | R37GM032443 | Sanford I Bernstein |
| National Institutes of Health | R01HL124091 | Anthony Cammarato |
| National Institutes of Health | R01AR064274 | Douglas M Swank |
| Rees-Stealy Research Foundation | Graduate Student Fellowship | Adriana S Trujillo |

The content is solely the responsibility of the authors and does not necessarily represent the official views of the National Institutes of Health.

## Author contributions

William A Kronert, Conceptualization, Formal analysis, Investigation, Writing—original draft, Writing—review and editing; Kaylyn M Bell, Data curation, Formal analysis, Investigation, Writing—original draft, Writing—review and editing; Meera C Viswanathan, Anju Melkani, Data curation, Formal analysis, Investigation, Writing—original draft; Girish C Melkani, Supervision, Investigation, Writing—original draft, Writing—review and editing; Adriana S Trujillo, Data curation, Formal analysis, Investigation, Methodology, Writing—original draft, Writing—review and editing; Alice Huang, Data curation, Formal analysis, Investigation; Anthony Cammarato, Sanford I Bernstein, Conceptualization, Formal analysis, Supervision, Funding acquisition, Writing—original draft, Project administration, Writing—review and editing; Douglas M Swank, Conceptualization, Data curation, Formal analysis, Supervision, Funding acquisition, Writing—original draft, Project administration, Writing—review and editing

## Author ORCIDs

Adriana S Trujillo (iD) http://orcid.org/0000-0002-0030-5715
Sanford I Bernstein (iD) http://orcid.org/0000-0001-7094-5390

## Decision letter and Author response

Decision letter https://doi.org/10.7554/eLife.38064.024
Author response https://doi.org/10.7554/eLife.38064.025

## Additional files

### Supplementary files

• Supplementary file 1. Power generation measured using the work loop technique under the same muscle length (ML) and frequency parameters. All three fiber types were oscillated through ten 0.75% ML amplitude and 125 Hz contraction cycles. These parameters are the optimal power producing parameters for most of the control fibers (average results are shown in *Table 2*). Means ± S.E.M are reported. N = 12 for the control and homozygous fibers, 13 for heterozygous fibers. Student's t-test with p<0.05 significantly different from control (*p<0.05, **p<0.01, ***p<0.001) or heterozygote (ˣp<0.05, ˣˣp<0.01, ˣˣˣp<0.001). Full genotypes are shown in parentheses: *PwMhc2* control (P{PwMhc2}/P{PwMhc2}; $Mhc^{10}/Mhc^{10}$); *R146N-15/+* heterozygote ($Mhc^{10}/+$; P{R146N-15}); *R146N-15* homozygote ($Mhc^{10}/Mhc^{10}$; P{R146N-15}/P{R146N-15}).
DOI: https://doi.org/10.7554/eLife.38064.018

• Supplementary file 2. Flight ability of *R146N* homozygous adults at two and seven days. Adult female flies were aged for two or seven days prior to flight testing. Transgenic flies were then assayed for the ability to fly up (U), horizontal (H), down (D) or not at all (N). Flight index = 6 U/T+ 4 H/T+ 2 D/T+ 0 N/T; T is the total number of flies tested, listed in parentheses. Flight index is mean ±S.E.M. Student's t-test with p<0.05 significantly different from same age *PwMhc2* (*p<0.05, **p<0.01, ***p<0.001) or from same fly line at 2 days of age (ˣp<0.05, ˣˣp<0.01, ˣˣˣp<0.001). Full genotypes are shown in parentheses: *PwMhc2* (P{PwMhc2}/P{PwMhc2}; $Mhc^{10}/Mhc^{10}$); *R146N* ($Mhc^{10}/Mhc^{10}$; P{R146N}/P{R146N}).
DOI: https://doi.org/10.7554/eLife.38064.019

• Supplementary file 3. Cardiomyocyte thickness measurements. At least three hearts were analyzed per line and genotype, including three sections along the anterior-poster axis at least 10 microns apart per sample, and ≥3 images each for dorsal or ventral areas per section analyzed. Means ± S.E. M are reported. A one-way ANOVA with the Bonferroni correction determined that no statistically significant differences (p<0.05) exist for any of the comparisons within or between samples at the same or different ages. Full genotypes are shown in parentheses: *PwMhc2/+* (P{PwMhc2}; $Mhc^{1}/+$); *R146N/+* ($Mhc^{1}/+$; P{R146N}).
DOI: https://doi.org/10.7554/eLife.38064.020

• Supplementary file 4. Cardiac parameters for *R146N-15/+ and R146N-28/+* mutant lines relative to *PwMhc2/+* control. Means ± S.E.M are reported. Two-way ANOVA results summarizing the statistical differences between the control and mutant flies can be found in *Figure 7*. No significant differences (p<0.05) in any cardiac physiological indices reported were noted between the two mutant lines via two-way ANOVA. Full genotypes are shown in parentheses: *PwMhc2/+* (P{PwMhc2}; $Mhc^{1}/+$); *R146N/+* ($Mhc^{1}/+$; P{R146N}).
DOI: https://doi.org/10.7554/eLife.38064.021

• Transparent reporting form
DOI: https://doi.org/10.7554/eLife.38064.022

### Data availability

Data generated or analysed during this study are included in the manuscript and supporting files.

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
