## [Decision Letter]

Thank you for submitting your article "Prolonged cross-bridge binding triggers muscle dysfunction in a fly model of myosin-based hypertrophic cardiomyopathy" for consideration by *eLife*. Your article has been reviewed by Didier Stainier as the Senior Editor, a Reviewing Editor, and three reviewers. The following individuals involved in review of your submission have agreed to reveal their identity: Lee Sweeney (Reviewer #1); Justin E Molloy (Reviewer #2).

The reviewers have discussed the reviews with one another and the Reviewing Editor has drafted this decision to help you prepare a revised submission.

Summary:

Hypertrophic cardiomyopathy (HCM) is the most common genetic heart condition involving abnormal heart structures and dysregulated cardiac functions. Mutations linked to HCM have been found in genes encoding proteins responsible for cardiac muscle contractility such as cardiac muscle myosins and troponins. However, it is not well understood how these mutations affect the encoded proteins and consequently cause HCM.

In this manuscript, Kronert et al., report an integrative study using a *Drosophila* mutant model on an HCM mutation, R146N, in the human cardiac myosin heavy chain gene (K146N is the *Drosophila* equivalent), to investigate a potential molecular mechanism underlying HCM. Structurally, the 146N mutation abolishes a salt bridge between the motor domain and the lever arm and presumably destabilizes the pre-power stroke state pf the myosin head. Biochemically, the 146N mutation enhanced ATPase activity of myosin and decreased actin filament sliding. Kinetically, 146N mutant muscles exhibited reduced work and power output. The biochemical and kinetics data suggests that the mutant myosin spends a longer time in strongly bound cross-bridge states. Degeneration in skeletal and cardiac muscles and decrease in flight ability occurred in 146N mutant flies in an age-dependent manner.

In *Drosophila* muscle, the 146N mutation appears to model the key aspects of a human mutation that has been suggested to interfere with interactions of cross-bridge packing rather than cross-bridge kinetics. The impact of such mutations cannot be properly assessed in solution and thus *Drosophila* muscles may provide an important model system for determining the impact of human HCM-causing mutations.

Essential revisions:

The three reviewers are all enthusiastic about the work, which they feel is of the highest quality and should be published in *eLife*. However, the comments of the reviewers should be addressed prior to publication. This will improve the readability and impact of the paper for a general audience. For the most part, this is more a matter of presentation and discussion, and not a need for further experimentation, so the authors should be able to accomplish this quickly. There is a request for additional modeling and structural information, which may or may not have already been done but not presented.

1) The significance the work is the fact that the impact of the mutation might have been restricted to cardiac muscle of vertebrates, but it is not. Not only does this allow for the use as *Drosophila* as a model organism to assess human HCM mutations, but it begs a broader discussion of the mechanism. The authors model the potential electrostatic interactions made by R146N and the lever arm and postulate that loss of this interaction might destabilize the pre-power stroke (PPS) state, which is supported by an increase in the actin-independent ATPase activity. However, they stop short in connecting this concept to the potential importance of this state for the myosin head packing during relaxation, which has been termed the super-relaxed state. There seems to be conservation of the head packing interactions, which were first described for single molecules of regulated smooth muscle myosin and has since been shown to fit the head-head interactions of myosin in filaments from tarantula muscle as well as in vertebrate cardiac muscle. This conservation of the packing is striking and must be the basis of what is happening in the flies as well. It has been modeled, and data confirms, that the PPS state is the state in which the myosin heads interact and pack, so anything that destabilizes the PPS conformation would be expected to disrupt the packing of the heads on the filament. This point is not stressed and yet is the real take home message.

2) The "myosin mesa" hypothesis really refers to the idea that for a large subset of the HCM mutations, the mutation does not directly alter the kinetics of the head, but instead disrupts the filament packing of the heads, which leads to effects such as reported for the R146N mutation of this paper. This should be emphasized, and some modeling of the impact of the R146N mutation on the packing of cardiac myosin heads into the filament would have been helpful. This could be added to Figure 1, but at the very least, further discussion should be added.

3) *R146N/+* heterozygous flies displayed myofibrillar discontinuity suggesting a dominant effect of this mutation on cardiac ultrastructure. The IFM from *R146N/+* heterozygotes also exhibited changes in power and work generation and cross-bridge kinetics. However, the authors did not present the IFM structure and the flight ability of the heterozygous flies. Is there an age-related delay in structural disruptions, since IFM myofibrils were deteriorated in older flies (7 day adults)? The authors suggested that this deterioration results from mechanical stress during muscle use. To strengthen this conclusion, transgenic flies of the same ages without muscle use could be examined. If there is no defect in the structure and the function of IFM (somatic muscle) in the heterozygotes at the ages presented, in contrast to the cardiac muscle, please discuss what properties of the two muscles can cause this difference. This is a significant point if the authors are using somatic muscles to investigate a cardiac muscle mutation.

4) The R146N mutation recapitulated some pathological phenomena of HCM such as disorganized myofibrils and diastolic dysfunction in *Drosophila*. However, the authors did not observe the hypertrophic effect on skeletal and cardiac muscles. The authors discussed that it might be consistent to the 'significant variability in the presence and degree of hypertrophy detected in patients carry the 146N mutation'. Since hypertrophy in *Drosophila* heart can occur due to abnormal cell proliferation from misregulated EGFR-Ras-Raf signaling (Yu et al., 2013), it would be interesting to test if increased stress by 146N mutation on *Drosophila* heart has an additive or synergistic effect on hypertrophy caused by abnormal RTK signaling.

---

## [Author Response]

Essential revisions:1) The significance the work is the fact that the impact of the mutation might have been restricted to cardiac muscle of vertebrates, but it is not. Not only does this allow for the use as Drosophila as a model organism to assess human HCM mutations, but it begs a broader discussion of the mechanism. The authors model the potential electrostatic interactions made by R146N and the lever arm and postulate that loss of this interaction might destabilize the pre-power stroke (PPS) state, which is supported by an increase in the actin-independent ATPase activity. However, they stop short in connecting this concept to the potential importance of this state for the myosin head packing during relaxation, which has been termed the super-relaxed state. There seems to be conservation of the head packing interactions, which were first described for single molecules of regulated smooth muscle myosin and has since been shown to fit the head-head interactions of myosin in filaments from tarantula muscle as well as in vertebrate cardiac muscle. This conservation of the packing is striking and must be the basis of what is happening in the flies as well. It has been modeled, and data confirms, that the PPS state is the state in which the myosin heads interact and pack, so anything that destabilizes the PPS conformation would be expected to disrupt the packing of the heads on the filament. This point is not stressed and yet is the real take home message.

This is an excellent point and certainly worthy of better discussion in the manuscript. We have added a paragraph to the Discussion section that introduces the super-relaxed state (SRX). We indicate how the interacting head motif appears to be the basis for SRX. We then state that the pre-power stroke state is found in the SRX and hence its likely disruption from the R146N mutation could enhance cross-bridge binding, leading to the phenotypes we have documented.

2) The "myosin mesa" hypothesis really refers to the idea that for a large subset of the HCM mutations, the mutation does not directly alter the kinetics of the head, but instead disrupts the filament packing of the heads, which leads to effects such as reported for the R146N mutation of this paper. This should be emphasized, and some modeling of the impact of the R146N mutation on the packing of cardiac myosin heads into the filament would have been helpful. This could be added to Figure 1, but at the very least, further discussion should be added.

As stated in #1, a paragraph has been added to the Discussion section on this topic has been added. We point out that disruption of filament packing could be a key element in the effect of the R146N mutation. Further, we indicate that we have recently shown (with Roger Craig) that *Drosophila* myosins are capable of forming the interacting head motif (Lee et al., 2018). However, the IHM and SRX have not yet been documented in *Drosophila* muscle and the observed phenotypes may be explained without invoking disruption of SRX. So, while we provided the requested further discussion, we felt that modeling of filament packing would be premature.

3) R146N/+ heterozygous flies displayed myofibrillar discontinuity suggesting a dominant effect of this mutation on cardiac ultrastructure. The IFM from R146N/+ heterozygotes also exhibited changes in power and work generation and cross-bridge kinetics. However, the authors did not present the IFM structure and the flight ability of the heterozygous flies. Is there an age-related delay in structural disruptions, since IFM myofibrils were deteriorated in older flies (7 day adults)? The authors suggested that this deterioration results from mechanical stress during muscle use. To strengthen this conclusion, transgenic flies of the same ages without muscle use could be examined. If there is no defect in the structure and the function of IFM (somatic muscle) in the heterozygotes at the ages presented, in contrast to the cardiac muscle, please discuss what properties of the two muscles can cause this difference. This is a significant point if the authors are using somatic muscles to investigate a cardiac muscle mutation.

The flight ability of heterozygotes is presented in Table 4 of the manuscript. We found a statistically significant reduction in flight ability at room temperature in two-day-old heterozygotes compared to controls (now mentioned in the Results). We now include a developmental series of electron micrographs as Figure 3—figure supplement 1. Here we show that assembly and structure are normal in the late pupa and two-hour-old adult, but there is minor myofibril deterioration at two days. While we have not assessed older heterozygous adults or determined whether muscle disuse prevents deterioration, it is clear that muscle function and structure (at least to some extent) are compromised in a dominant fashion, as is the case in the heart.

4) The R146N mutation recapitulated some pathological phenomena of HCM such as disorganized myofibrils and diastolic dysfunction in Drosophila. However, the authors did not observe the hypertrophic effect on skeletal and cardiac muscles. The authors discussed that it might be consistent to the 'significant variability in the presence and degree of hypertrophy detected in patients carry the 146N mutation'. Since hypertrophy in Drosophila heart can occur due to abnormal cell proliferation from misregulated EGFR-Ras-Raf signaling (Yu et al., 2013), it would be interesting to test if increased stress by 146N mutation on Drosophila heart has an additive or synergistic effect on hypertrophy caused by abnormal RTK signaling.

While a hypertrophic response in the *Drosophila* cardiac tube was not resolved in our studies, our primary goal was to uncover the mechanical effects of a human HCM-causing mutation in myosin across the molecular, myofibrillar, cellular, and tissue scales and, thereby, provide key insights into defining the mechanism behind K146N myosin-based cardiomyopathy. We believe we accomplished this goal. We would like to emphasize that Yu et al., (2013) reported that hypertrophic cardiomyopathy in fly hearts was characterized by decreased end diastolic lumen dimensions that resulted from abnormal cardiomyocyte fiber morphology and increased heart wall thickness, rather than abnormal cell proliferation. According to the authors there were no changes in cardiomyocyte cell numbers. Nonetheless, we agree with the reviewer that it would be interesting to determine the effects of the myosin mutation in the context of abnormal receptor tyrosine kinase signaling. In fact, such genetic interaction experiments are perfectly suited for the fly model. However, we feel these experiments are non-trivial to perform and are well beyond the scope of the current manuscript.